# Unified Privacy Guarantees for Decentralized Learning via Matrix Factorization

**Aurélien Bellet**
Inria, Université de Montpellier, INSERM
Montpellier, France

**Edwige Cyffers**
Institute of Science & Technology Austria (ISTA)
Klosterneuburg, Austria

**Davide Frey, Romaric Gaudel, Dimitri Lerévérend,**[*] **François Taïani**
University of Rennes, Inria, CNRS, IRISA
Rennes, France

## Abstract

Decentralized Learning (DL) enables users to collaboratively train models without sharing raw data by iteratively averaging local updates with neighbors in a network graph. This setting is increasingly popular for its scalability and its ability to keep data local under user control. Strong privacy guarantees in DL are typically achieved through Differential Privacy (DP), with results showing that DL can even amplify privacy by disseminating noise across peer-to-peer communications. Yet in practice, the observed privacy-utility trade-off often appears worse than in centralized training, which may be due to limitations in current DP accounting methods for DL. In this paper, we show that recent advances in centralized DP accounting based on Matrix Factorization (MF) for analyzing temporal noise correlations can also be leveraged in DL. By generalizing existing MF results, we show how to cast both standard DL algorithms and common trust models into a unified formulation. This yields tighter privacy accounting for existing DP-DL algorithms and provides a principled way to develop new ones. To demonstrate the approach, we introduce MAFALDA-SGD, a gossip-based DL algorithm with user-level correlated noise that outperforms existing methods on synthetic and real-world graphs.

## 1 Introduction

In Decentralized Learning (DL), participants collaboratively train a shared model by exchanging model parameters on a peer-to-peer communication graph (Lian et al., 2017; 2018; Koloskova et al., 2020; Beltrán et al., 2022; Tian et al., 2023; Yuan et al., 2024). By removing the central server, DL offers faster deployment, higher scalability, and improved robustness. Since data remains local and each participant sees only a subset of communications, DL is also motivated by privacy concerns (Cyffers & Bellet, 2022; Cyffers et al., 2022; Kairouz et al., 2021c). However, decentralization alone is insufficient to ensure privacy, as exchanged messages can still leak sensitive information that enables inference or reconstruction of local data (El Mrini et al., 2024; Pasquini et al., 2023). Data protection thus requires additional mechanisms beyond decentralization.

Differential Privacy (DP) (Dwork et al., 2006) is the gold standard for privacy in machine learning. DP protects participation by ensuring that outputs do not depend too strongly on any single data point, typically through noise injection. Extending DP to Decentralized Learning is challenging: unlike in centralized settings, where a trusted curator releases only the final output, DL exposes intermediate peer-to-peer messages to participants. A natural option is to analyze DP-DL algorithms under Local DP (Kasiviswanathan et al., 2008; Bellet et al., 2018; Wang et al., 2015), which assumes

---

[*]Corresponding author, `dimitri.lereverend@inria.fr`

all messages are released publicly, but this overly conservative trust model often leads to poor privacy–utility trade-offs (Chan et al., 2012). More realistic models, such as Pairwise Network DP (PNDP) (Cyffers et al., 2022; Cyffers & Bellet, 2022) and Secret-based LDP (SecLDP) (Allouah et al., 2024), have recently been proposed to account for adversaries with only partial knowledge of the system, showing that decentralization can amplify privacy guarantees relative to Local DP.

Unfortunately, despite these recent advances, designing DP-DL algorithms and analyzing their privacy guarantees remains a difficult and technical endeavor, and existing results rely on ad hoc proofs tailored to specific algorithms and trust models (Cyffers et al., 2024; Biswas et al., 2025; Allouah et al., 2024), rather than on a general, principled approach. There is also room to obtain tighter privacy guarantees by more carefully accounting for correlated noise arising from redundant exchanges between nodes, both in parallel and across timesteps—an aspect largely overlooked in existing analyses, which can lead to overly pessimistic bounds.

In this work, we address these limitations by introducing a unified formulation for analyzing privacy guarantees in DP-DL, building on recent advances in centralized privacy accounting via the Matrix Factorization (MF) mechanism (Kairouz et al., 2021a; Pillutla et al., 2025). MF uses a well-chosen factorization of a workload matrix representing the algorithm to exploit noise correlations and achieve stronger privacy–utility trade-offs, but has been so far applied only in centralized contexts. We propose a framework that make MF applicable to ensuring DP in decentralized learning. Achieving this requires addressing several fundamental challenges. First, decentralized learning algorithms must be encoded as a single tractable matrix multiplication, a formulation that has not been achieved in prior work. Second, the diversity of trust models requires disentangling the matrix governing privacy guarantees from the one driving optimization schemes—two quantities that collapse into a single object in the centralized setting. Finally, incorporating these elements requires generalizing MF to richer classes of workload matrices and constraints. Our framework overcomes these challenges, providing a principled foundation to analyze existing DP-DL algorithms across different trust models and to design new ones. Specifically, our contributions are as follows:

 (i) We generalize existing privacy guarantees for MF to broader settings;
 (ii) We show how existing DP-DL algorithms and trust models can be analyzed as specific instances of our generalized MF framework;
(iii) We use our framework to design a new algorithm, MAFALDA-SGD; and
(iv) We evaluate our approach on synthetic and real-world graphs and datasets, demonstrating tighter privacy guarantees for existing algorithms and superior performance of MAFALDA-SGD compared to prior methods.

## 2 RELATED WORK

**Matrix factorization mechanism.** The Matrix Factorization (MF) mechanism for SGD first appeared as a generalization of correlated noise in the online setting (Kairouz et al., 2021b), based on tree mechanisms (Dwork et al., 2010; Jain et al., 2012), and was later formalized by Denisov et al. (2022). More recent work improves the factorization either by designing more efficient decompositions (Choquette-Choo et al., 2023), or by allowing additional constraints on the strategy matrix, i.e., the matrix encoding locally applied noise correlations on the gradient. Existing constraints typically enforce a limited window for temporal correlations via banded matrices, which reduces computation (Choquette-Choo et al., 2023; Kalinin & Lampert, 2024), whereas in this work our constraint in MAFALDA-SGD enforces that correlations occur only within nodes. Other works extend the mechanism to more complex participation schemes (Choquette-Choo et al., 2023). For a more comprehensive survey, we refer the reader to Pillutla et al. (2025).

**Differentially private decentralized learning.** Private decentralized learning was first studied under the trust model of Local DP (Bellet et al., 2018; Cheng et al., 2019; Huang et al., 2015), which suffers from poor utility due to overly pessimistic assumptions: participants are only protected by their own noise injection, since all messages are assumed to be known to the attacker. Relaxed trust models have since gained traction, such as Pairwise Network Differential Privacy (PNDP) (Cyffers et al., 2022), initially introduced for gossip averaging and later extended to several algorithms (Cyffers et al., 2024; Li et al., 2025; Biswas et al., 2025). PNDP assumes that the attacker is a node participating in the algorithm, and that it is thus sufficient to protect what a node observes during the execution—typically only the messages it sends or receives. Similarly, SecLDP (Allouah et al.,

2024) enforces DP conditional on a set of secrets that remain hidden from the attacker. The idea of correlating noise has a long history in DL, first as an obfuscation method against neighbors without formal guarantees (Mo & Murray, 2017; Gupta et al., 2020). More recent works provide DP guarantees for algorithms with correlated noise (Sabater et al., 2022; Allouah et al., 2024; Biswas et al., 2025), but correlations are not optimized and instead dictated by what can be proven, which explains why our method MAFALDA-SGD significantly outperforms them. Finally, the concept of representing attacker knowledge in DL via a matrix that can be constructed algorithmically was introduced by El Mrini et al. (2024) and more recently studied by Koskela & Kulkarni (2025), but only for specific cases of algorithms and trust models, and without establishing a connection to the matrix factorization mechanism.

## 3 BACKGROUND: MATRIX FACTORIZATION MECHANISM

In this section, we review the centralized version of Differentially Private Stochastic Gradient Descent (DP-SGD) and its correlated noise variant based on matrix factorization (MF-SGD) (Denisov et al., 2022), which we will later adapt to the decentralized case.

Differential Privacy (DP) (Dwork et al., 2006) ensures that the output of a mechanism does not reveal too much information about any individual data record in its input. A mechanism satisfies DP if its output distribution remains close on any two neighboring datasets $\mathcal{D}$ and $\mathcal{D}'$ that differ in one record (denoted $\mathcal{D} \simeq \mathcal{D}'$). Although DP is classically defined in terms of $(\epsilon, \delta)$-DP, in this paper we adopt Gaussian DP (GDP), which is particularly well suited to mechanisms with Gaussian noise. GDP offers tighter privacy accounting, simpler analysis, and can be converted back into Rényi DP or $(\epsilon, \delta)$-DP (Dong et al., 2022).

**Definition 1** (Gaussian Differential Privacy (GDP) — Dong et al. 2022; Pillutla et al. 2025). *A randomized mechanism $\mathcal{M}$ satisfies $\mu$-Gaussian Differential Privacy ($\mu$-GDP) if, for any neighboring datasets $\mathcal{D} \simeq \mathcal{D}'$, there exists a (possibly randomized) function $h$ such that*

$$h(Z) \stackrel{d}{=} \mathcal{M}(\mathcal{D}), \quad Z \sim \mathcal{N}(0,1), \qquad h(Z') \stackrel{d}{=} \mathcal{M}(\mathcal{D}'), \quad Z' \sim \mathcal{N}(\mu, 1),$$

*where $\stackrel{d}{=}$ denotes equality in distribution.*

In practice, Gaussian DP is obtained by adding appropriately calibrated Gaussian noise, which is the central building block of the centralized DP-SGD algorithm. Given a training dataset $\mathcal{D}$, a learning rate $\eta > 0$, and starting from an initial model $\theta_0 \in \mathbb{R}^d$, the DP-SGD update can be written as:

$$\theta_{t+1} = \theta_t - \eta\,(g_t + z_t), \quad g_t = \text{clip}\,(\nabla f(\theta_t, \xi_t), \Delta), \quad z_t \sim \mathcal{N}(0, \Delta^2 \sigma^2 I_d), \tag{1}$$

with $\xi_t$ a sample from $\mathcal{D}$ and $\Delta = \sup_{\mathcal{D} \simeq \mathcal{D}'} \|\nabla f(\mathcal{D}) - \nabla f(\mathcal{D}')\|_2$ the sensitivity of $\nabla f$ enforced by the clipping operation. Even for this simple version of DP-SGD without correlated noise, we can define a so-called workload matrix $A^{\text{pre}} \in \mathbb{R}^{T \times T}$ with $A_{ij}^{\text{pre}} = 1_{i \geq j}$, such that:

$$\theta = 1_T \otimes \theta_0 - (A^{\text{pre}} G + Z), \quad G \in \mathbb{R}^{T \times d}, \tag{2}$$

where $\theta$ is the matrix of stacked $\theta_1, ..., \theta_T$, $G$ is the matrix of stacked gradients and $Z \sim \mathcal{N}(0, \sigma^2 \Delta)^{T \times d}$. Here $G$ corresponds to the data-dependent part of the algorithm, and $\theta$ corresponds to the outputs of DP-SGD. We thus aim to protect $G$ as much as possible by adding a large variance vector $Z$, while still achieving the best possible final $\theta_T$, typically by keeping $Z$ as small as possible.

To reach these contradictory goals, the matrix factorization mechanism exploits the postprocessing property of DP—which ensures that if a mechanism $\mathcal{M}$ is $\mu$-GDP, then for every function $f$, $f \circ \mathcal{M}$ is also $\mu$-GDP—to allow noise correlation across iterations and improve the privacy–utility trade-off. For any factorization $A^{\text{pre}} = BC$, one can rewrite $A^{\text{pre}} G + BZ$ as $A^{\text{pre}}(G + C^\dagger Z)$, where the multiplication by $A^{\text{pre}}$ is seen as a post-processing of the term $(G + C^\dagger Z)$ and $C^\dagger$ is the pseudo-inverse of $C$. This factorization leads to a slight modification of DP-SGD, resulting in the MF-SGD updates:

$$\theta_{t+1} = \theta_t - \eta\big(g_t + \sum_{\tau=0}^{t} C_{t,\tau}^\dagger z_\tau\big). \tag{3}$$

To derive DP guarantees for MF-SGD, it suffices to analyze the privacy guarantees of $(G + C^\dagger Z)$. For neighboring datasets $\mathcal{D} \simeq \mathcal{D}'$ differing only in one record $x$, differences in $G$ occur whenever $x$ is used, which may happen at multiple timesteps if the dataset is cycled through several times. The

---

**Algorithm 1** MF-D-SGD: Matrix Factorization Decentralized Stochastic Gradient Descent

---

**Require:** $W \in \mathbb{R}^{n \times n}, C, T, \Delta_g, \sigma, \theta_0 \in \mathbb{R}^{n \times d}, Z \sim \mathcal{N}\left(0, \Delta_g^2 \sigma^2\right)^{nT \times d}$
1: **for all** node $u$ **in parallel do**
2:     **for** $t = 1$ **to** $T$ **do**
3:         $g_t^{(u)} \leftarrow \text{clip}\left[\nabla f_u(\theta_t^{(u)}, \xi_t^{(u)}), \Delta_g\right]$ with $\xi_t^{(u)} \sim \mathcal{D}_u$        ▷ Clipped gradient
4:         $\theta_{t+\frac{1}{2}}^{(u)} \leftarrow \theta_t^{(u)} - \eta\big(g_t^{(u)} + (C^\dagger Z)_{[nt+u]}\big)$        ▷ Local update
5:         Send $\theta_{t+\frac{1}{2}}^{(u)}$ to all neighbors $v \in \Gamma_u$
6:         Receive $\theta_{t+\frac{1}{2}}^{(v)}$ from all neighbors $v \in \Gamma_u$
7:         $\theta_{t+1}^{(u)} \leftarrow \sum_{v \in \Gamma_u} W_{[u,v]} \theta_{t+\frac{1}{2}}^{(v)}$        ▷ Local average
8: **return** $\theta_{T+1}^{(u)}, \forall u \in [\![1, n]\!]$

---

*participation scheme* $\Pi$ is the set of all *participation patterns* $\pi$, encoding whether $x$ participates in a given row of $G$ (see Chapter 3 in Pillutla et al., 2025). This extends the neighboring relation from datasets to gradients: $G \simeq_\Pi G'$. The sensitivity is then the worst case over all participation patterns:

$$\text{sens}_\Pi(C) := \text{sens}_\Pi(G \mapsto CG) = \max_{G \simeq_\Pi G'} \|C(G - G')\|_F. \tag{4}$$

For a fixed sensitivity $\text{sens}_\Pi(C)$, generating $Z \sim \mathcal{N}(0, \sigma^2 \text{sens}_\Pi(C)^2)^{T \times d}$ ensures that $(G + C^\dagger Z)$ is $\frac{1}{\sigma}$-GDP. Optimizing the noise correlation is typically done by minimizing $\text{sens}(C)^2 \|B\|^2$ under the constraint $A^{\text{pre}} = BC$, where $\|B\|^2$ represents the utility of the factorization. While the privacy analysis is straightforward if $G$ and $G'$ are fixed in advance, the guarantee remains valid when $G$ is computed *adaptively* through time, with $g_t$ depending on $g_1, \ldots, g_{t-1}$. This has been proven only for workload matrices that are square, lower triangular, and full rank (Denisov et al., 2022; Pillutla et al., 2025). We show in Section 5 that the privacy guarantees also hold in a more general case, thereby enabling adaptivity in DL as well.

## 4 DECENTRALIZED LEARNING AS A MATRIX FACTORIZATION MECHANISM

In this section, we show that many DL algorithms can be cast as instances of the matrix factorization mechanism. To do so, we must address several challenges: encoding the updates of the algorithms, representing the sensitive information, and capturing what is known to a given attacker in a form suitable for analyzing DP guarantees. For concreteness, we first show in Section 4.1 how to proceed for a specific yet already fairly general algorithm, which we call MF-D-SGD, under Local DP, and then further generalize to a large class of DL algorithms and trust models in Section 4.2.

### 4.1 WARM-UP: MF-D-SGD UNDER LOCAL DP

We introduce the decentralized setting studied in our work and build step-by-step the encoding of a Decentralized SGD (D-SGD) algorithm where DP is introduced in a similar way as in MF-SGD. We refer to this algorithm as MF-D-SGD and present its pseudocode from the users' perspective in Algorithm 1 and from the matrix perspective in Figure 1.

We consider that $n$ users, represented as nodes of a graph $\mathcal{G} = (\mathcal{V}, \mathcal{E})$, collaboratively minimizing a loss function by alternating gradient steps computed on their local datasets $(\mathcal{D}_u)_{u \in V}$ (Lines 3 and 4) and gossiping steps, i.e., averaging their local models with their neighbors (Line 5–7). In DL, two datasets $\mathcal{D} = (\mathcal{D}_u)_{u \in V}$ and $\mathcal{D}' = (\mathcal{D}'_u)_{u \in V}$ are neighboring if they differ in exactly one record at a single node $u$, i.e., $\mathcal{D}_v = \mathcal{D}'_v$ for all $v \neq u$ and $\mathcal{D}'_u = \mathcal{D}_u \setminus \{x\} \cup \{x'\}$. We write this as $\mathcal{D} \simeq_u \mathcal{D}'$.

**Model updates.** We stack the local parameters of the $n$ nodes as a vector $\theta_t \in \mathbb{R}^{n \times d}$, where $\theta_t^{(u)}$ is the local parameter of node $u$ at time $t \in [\![0, T-1]\!]$. The gradient step can then be written as $\theta_{t+\frac{1}{2}} = \theta_t - \eta\left(G_t + C_t^\dagger Z\right)$, where each row $u$ of $G_t$ is the vector $G_t^{(u)} = \nabla f_u\big(\theta_t^{(u)}, \xi_t^{(u)}\big)^\top$, and $C_t^\dagger \in \mathbb{R}^{n \times m}$ is the block of $C^\dagger$ from the $(tn+1)$-th row to the $(t+1)n$-th one. $C_t^\dagger Z$ captures arbitrary noise correlation, and setting $C = I_{nT}$ recovers the local updates of DP-D-SGD, the simple

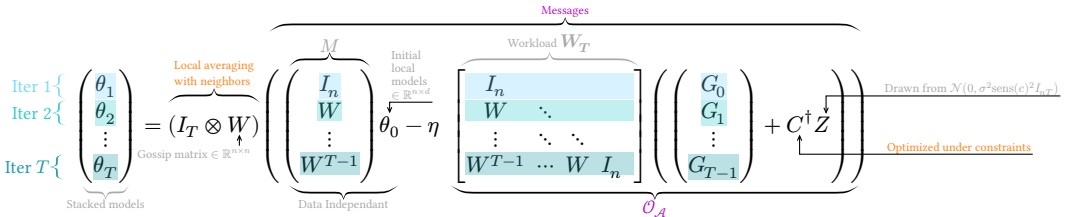

Figure 1: Overview of the MF-D-SGD algorithm written as a single equation.

decentralized SGD with uncorrelated noise. After the local updates, nodes perform a gossiping step, exchanging their models with neighbors and averaging the intermediate models $\theta_{t+\frac{1}{2}}$ using the gossip matrix $W$.

**Definition 2** (Gossip matrix). *A gossip matrix $W \in \mathbb{R}_{\geq 0}^{n \times n}$ on the graph $G = (\mathcal{V}, \mathcal{E})$ is a stochastic matrix, i.e., $\sum_{v=1}^{n} W_{uv} = 1$, and if $W_{uv} > 0$ then $v$ is a neighbor of $u$, denoted $v \in \Gamma_u$.*

With this definition, the gossiping step can be written as $\theta_{t+1} = W\theta_{t+\frac{1}{2}}$. Thus, one full iteration of MF-D-SGD corresponds to $\theta_{t+1} = W\left(\theta_t - \eta\left(G_t + C_t^\dagger Z\right)\right)$.

We now stack all steps of MF-D-SGD over $T$ iterations. This requires defining a matrix $M$ with stacked powers of $W$ (starting with power 0) and a larger matrix $\mathbf{W}_T \in \mathbb{R}^{nT \times nT}$ with blocks of size $n \times n$ in lower triangular Toeplitz form, where the main diagonal is filled with $I_n$ and the $i$-th diagonal with $W^{i-1}$. The whole algorithm can then be summarized as:

$$\theta = (I_T \otimes W)\left(M\theta_0 - \eta\,\mathbf{W}_T\left(G + C^\dagger Z\right)\right), \tag{5}$$

where $\theta$ and $G$ denote the collections of local models and gradients, respectively, after $T$ iterations.

**Remark 3.** *Up to this point, we have assumed that the gossip matrix $W$ is fixed across iterations for simplicity's sake (notably when explaining Figure 1). In the remainder of the paper we move to the more general case of time-varying gossip matrices $W_t$. This change only requires replacing powers of a single matrix gossip $W$ by products of matrices in the expression of $\mathbf{W}_T$. See Appendix A.7 for details of this more general expression.*

**Attacker knowledge.** A subtlety arises from the fact that under decentralized trust models, an attacker typically does not observe the sequence of local models $\theta$ directly. Hence, we should not apply the MF mechanism to Equation (5), but rather the peer-to-peer messages. Under the Local DP trust model, where all messages exchanged by the nodes throughout the algorithm are assumed to be public, the view $\mathcal{O}_A$ of the attacker can be seen as a MF mechanism:

$$\mathcal{O}_A := \mathbf{W}_T\left(G + C^\dagger Z\right). \tag{6}$$

## 4.2 GENERAL FORMULATION

The factorization obtained in previous section for MF-D-SGD requires two assumptions: (i) at each iteration, every node sends the same message to all of its neighbors, and (ii) the attacker observes all messages shared on the network. To encompass a broader range of algorithms and trust models, this section proposes more general definitions for both. To the best of our knowledge, these definitions cover all existing DP-DL algorithms and trust models.

**Definition 4** (Linear DL algorithm). *A decentralized learning algorithm is* linear *if all observable quantities can be expressed as a linear combination of the concatenated gradients $G$ and the concatenated noise $Z$.*

This definition covers the Zip-DL algorithm, in which a user $u$ applies distinct yet correlated noises to its communications within the same gossiping step (Biswas et al., 2025). We complete our framework by defining trust models on linear DL algorithms as follows.

**Definition 5** (Attacker knowledge). *Given a Linear DL algorithm, the* attacker knowledge *of a trust model, i.e., the information the attacker $\mathcal{A}$ derives from its observations, can be expressed as $\mathcal{O}_A := AG + BZ$.*

This definition encompasses not only scenarios in which knowledge is obtained by observing messages or models, but also situations where the attacker sees only a subset of them, as in PNDP. It further allows encoding: (i) gradients already known to the attacker, e.g., when a user knows its own gradients or those of colluding users (in which case the corresponding columns of $A$ are 0); (ii) information available about the noise, e.g., when a user knows the noise it added, or in SecLDP when some neighbors' noise values are known (in which case the corresponding columns of $B$ are 0), as shown by the following theorem.

Remarkably, these general definitions not only capture all widely used trust models and algorithms, but the existence of a corresponding factorization for these cases is also guaranteed, as established by the following theorem.

**Theorem 6.** *For each of the three trust models—LDP, PNDP, and SecLDP—and for all existing DL algorithms, there exist matrices $A$, $B$, and $C$ such that the attacker knowledge can be expressed as $\mathcal{O}_\mathcal{A} = AG + BZ$ with $A = BC$.*

We refer the reader to Appendix A for the proof, where we construct the matrices $A$, $B$ and $C$ for each trust model based on the algorithms that introduced them. These matrices are constructed in two steps: (i) forming three distinct blocks corresponding to observed messages, known gradients, and known noise; (ii) removing from the overall matrix $A$ the contribution of the known gradients. The first step ensures that all known information is properly accounted for, while the second step guarantees the existence of a factorization $A = BC$, which is crucial to obtain GDP guarantees, as established in the next section by Theorem 8.

## 5   MORE GENERAL PRIVACY GUARANTEES FOR MATRIX FACTORIZATION

In Section 4, we showed how to cast the attacker knowledge in DL as a MF mechanism $AG + BZ$ with $A = BC$. However, the resulting matrices do not satisfy the assumptions used in existing MF results, which require $A$ to be a square, full rank and lower-triangular matrix. In this section, we extend the MF mechanism's differential privacy guarantees to workloads $A$ that may violate these assumptions. To do so, we first introduce a more precise definition of *sensitivity*, which is fundamental for evaluating the quality of a factorization (Pillutla et al., 2025).

**Definition 7** (Sensitivity under participation, generalized). *Let $B$ and $C$ be two matrices. The sensitivity of $C$ with respect to $B$ and a participation scheme $\Pi$ is*

$$\operatorname{sens}_\Pi(C; B) := \max_{G \simeq_\Pi G'} \left\| C\left(G - G'\right) \right\|_{B^\dagger B}, \tag{7}$$

*with $\|A\|_B^2 := \operatorname{tr}\left(A^\top B A\right)$.*

With this definition, we state our main privacy theorem, whose proof is provided in Appendix B.

**Theorem 8.** *Let $\mathcal{O}_\mathcal{A} = AG + BZ$ be the attacker knowledge of a trust model on a linear DL algorithm, and denote $\mathcal{M}(G)$ the corresponding mechanism. Let $\Pi$ be a participation scheme for $G$. For $Z \sim \mathcal{N}\left(0, \nu^2\right)^{m \times d}$, when $A$ is a column-echelon matrix and there exists some matrix $C$ such that $A = BC$ with*

$$\nu = \sigma \operatorname{sens}_\Pi(C; B) \quad with \quad \operatorname{sens}_\Pi(C; B) \leq \max_{\pi \in \Pi} \sum_{s,t \in \pi} \left| \left(C^\top B^\dagger B C\right)_{s,t} \right|, \tag{8}$$

*then $\mathcal{M}$ is $\frac{1}{\sigma}$-GDP, even when $G$ is chosen adaptively.*

Theorem 8 shows that the privacy guarantees of a matrix factorization mechanism remain valid for a broader class of matrices $A$, even when the gradient vector is chosen *adaptively*, i.e., when it may depend arbitrarily on all *past* information (Denisov et al., 2022). The column-echelon form property of the matrix ensures it cannot depend on *future* information. This generalization of the usual lower-triangular property is necessary when considering attackers who have only partial information over the network. Most sets of observations that follow the natural causal ordering in DL can be rewritten as a column-echelon matrix.

Our result generalizes existing results by allowing $A$ to be rectangular and possibly rank-deficient, a relaxation crucial for applying it to DL algorithms. The price of this extension is a dependency on $B$:

since $B$ may not be invertible, $C$ is not unique, and the sensitivity now depends on both the encoder matrix $C$ and the decoder matrix $B$. The correction $B^\dagger B$ can be interpreted as a projection of the columns of $C$ onto the row space of $B$, discarding unobserved gradient combinations. When $B$ is square and full rank, as in prior work, we recover the known formula $\text{sens}_\Pi(C; B) = \text{sens}_\Pi(C; I) = \text{sens}_\Pi(C)$.

**Remark 9.** *In DL, the participation scheme $\Pi$ is also slightly modified. A participation pattern $\pi$ is associated to a local dataset $\mathcal{D}_u$ and can be non-negative only for the rows in $\{u, n + u, \ldots, (T - 1)n + u\}$, which corresponds to gradients computed from $\mathcal{D}_u$.*

**Remark 10.** *Following Corollary 1 of Dong et al. (2022), $\mu$-GDP directly and tightly translates into $(\epsilon, \delta)$-DP, with appropriate $\epsilon$ and $\delta$ values. Moreover, it is also possible to derive $(\alpha, \epsilon)$-Rényi DP guarantees Gil et al. (2013).*

## 6 A Novel Algorithm with Optimized Noise Correlation

In this section, we showcase how the formalization developed in the previous sections enables optimizing noise correlation in DL algorithms. Specifically, we introduce MAFALDA-SGD (MAtrix FActorization for Local Differentially privAte SGD), a new algorithm for DL with LDP, which can be seen as a special instance of MF-D-SGD.

We first define an objective function $\mathcal{L}_{\text{opti}}$ capturing the optimization error of MF-D-SGD.

**Definition 11.** *Consider Equation (5) with encoder matrix $C$, such that $\mathbf{W}_T = BC$. The optimal correlation can be found by minimizing the following objective function:*

$$\mathcal{L}_{opti}(\mathbf{W}_T, B, C) := \text{sens}_\Pi(C; B)^2 \left\| (I_T \otimes W) \mathbf{W}_T C^\dagger \right\|_F^2. \tag{9}$$

In DL, the optimization loss is defined with respect to the averaged model, as explicitly written in Equation (5) for MF-D-SGD. However, sensitivity must also account for the attacker knowledge (Theorem 8), which in this case corresponds to $\mathcal{O}_\mathcal{A}$ (defined in Equation (6)). The difference between the two terms explains why, despite having a workload matrix equal to $\mathbf{W}_T$, the second term involves $(I_T \otimes W) \mathbf{W}_T$ instead.

While one could attempt to minimize Equation (9) directly, doing so is impractical without additional constraints. The encoder matrix $C$ has size $nT \times nT$, which becomes intractable for large graphs or long time horizons. Furthermore, the resulting correlated noises could necessitate correlations across nodes, which in turn would require trust between them. These challenges motivate the introduction of structural constraints on $C$.

Here, we focus on Local DP guarantees, where nodes cannot share noise. Accordingly, we consider only *local correlations*, constraining $C = C_{\text{local}} \otimes I_n$ so that the block structure enforces locality and all nodes follow the same pattern, reducing the cost of computing $C$. For simplicity, we also assume that all nodes adhere to the same local participation pattern. Correlation schemes proposed for the centralized setting naturally fit this form; for example, AntiPGD (Koloskova et al., 2023) subtracts the noise added at a given step in the next step. However, we find that this correlation pattern performs poorly when used in DL (see Section 7), motivating our approach of optimizing correlations specifically for decentralized algorithms.

Under LDP and local noise correlation, sensitivity depends only on $C_{\text{local}}$. We simplify the general objective in Definition 11 with the following lemma, with its proof deferred to Appendix D.

**Lemma 12.** *Consider the LDP trust model ($\tilde{A} = \mathbf{W}_T$, $\tilde{B} = B$) and local noise correlation $C = C_{local} \otimes I_n$. Define:*

$$A_i := \left[ (I_T \otimes W) \mathbf{W}_T \mathbf{K}^{(T,n)} \right]_{[:, iT:(i+1)T-1]}, \quad H := \sum_{i=1}^n A_i^\top A_i, \tag{10}$$

*where $\mathbf{K}^{(T,n)}$ is a commutation matrix that commutes space and time in our representation (Loan, 2000). Then, the objective function of Equation (9) is equivalent to:*

$$\mathcal{L}_{opti}(\mathbf{W}_T, B, C_{local}) = \mathcal{L}_{opti}(\mathbf{W}_T, C_{local}) = \text{sens}_{\Pi_{local}}(C_{local})^2 \left\| L C_{local}^\dagger \right\|_F^2, \tag{11}$$

*where $L$ is lower triangular and obtained by Cholesky decomposition of $H = L^\top L$.*

---

**Algorithm 2** MAFALDA-SGD: MAtrix FActorization for Local Differentially privAte SGD

---

**Require:** $W \in \mathbb{R}^{n \times n}, T, \Delta_g, \sigma, \theta_0 \in \mathbb{R}^{n \times d}$
 1: Build workload matrix $\mathbf{W}_T$      ▷ Figure 1
 2: $H \leftarrow \sum_{i=1}^{n} A_i^\top A_i$ where $A_i := \left[(I_T \otimes W) \mathbf{W}_T \mathbf{K}^{(T,n)}\right]_{[:, iT:(i+1)T-1]}$      ▷ Lemma 12
 3: $L \leftarrow$ Cholesky decomposition of $H$      ▷ $H = L^\top L$
 4: $C_{\text{mafalda}} \leftarrow \min_{C_{\text{local}}} \mathcal{L}_{\text{opti}}(L, C_{\text{local}})$      ▷ Using L-BFGS Choquette-Choo et al. (2023)
 5: **return** MF-D-SGD($W, C_{\text{mafalda}}, T, \Delta_g, \sigma, \theta_0$)      ▷ Algorithm 1

---

This yields a standard factorization problem for $L$, which can be solved using existing frameworks for the MF mechanism Choquette-Choo et al. (2023); Pillutla et al. (2025). In practice, these frameworks operate on the Gram matrix $H = L^\top L$, avoiding the need for Cholesky decomposition.

Algorithm 2 summarizes our novel MAFALDA-SGD algorithm. It first computes the optimal correlation by optimizing for the privacy-utility trade-off in DL based on the objective function in Lemma 12, before running MF-D-SGD with the resulting $C_{\text{mafalda}}$. Note that $C_{\text{mafalda}}$ can be computed offline by any party that knows the gossip matrix and the participation pattern.

**Remark 13.** *The complexity of Algorithm 2 is driven by (i) the computation of the matrix $L$, which costs $\mathcal{O}(n^3 T + n^2 T^3)$ in time, and $\mathcal{O}(n^2 T^2)$ in space, and (ii) the minimization of $\mathcal{L}_{opti}$ which is $\mathcal{O}(T^3)$ per step of L-BFGS. Correlation restarts (Pillutla et al., 2025) can be used to keep $T$ within reasonable bounds.*

## 7 EXPERIMENTS

In this section, we highlight the advantages of our approach by first showing that we can achieve significantly tighter privacy accounting for an existing algorithm under PNDP (Section 7.1), and then demonstrating that MAFALDA-SGD outperforms existing DL algorithms under LDP. We use both synthetic and real-world graphs. Additional experiments are reported in Appendix C.

### 7.1 TIGHTER ACCOUNTING FOR PNDP

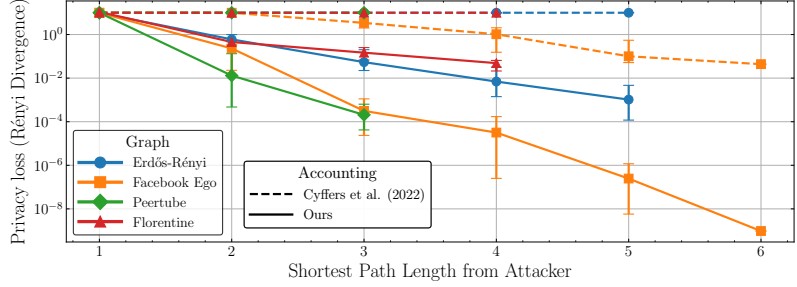

Figure 2: PNDP loss accounting for DP-D-SGD: comparison between the original accounting of Cyffers et al. (2022) (dashed lines) and our accounting (solid lines) under the PNDP trust model. We report the Rényi divergence (fixing $\alpha = 2$ w.l.o.g., since the results are proportional to $\alpha$) for the minimum, maximum, and average in error bars over nodes at a given distance.

In this experiment, we consider DP-D-SGD—which corresponds to the Muffliato-SGD algorithm from Cyffers et al. (2022) when the number of gossip steps is set to $K = 1$—and compute its PNDP guarantee with two different privacy accounting methods: the one proposed by Cyffers et al. (2022) and our own, obtained by casting the guarantees of Theorem 8 to the case of DP-D-SGD with a PNDP attacker as follows. For $C_a = I_{nT}$ and $B_a = \mathbf{P}_a \mathbf{W}_T$, using Theorem 8, the bound on sensitivity of DP-D-SGD under PNDP is

$$\operatorname*{sens}_{\Pi}(C_a; B_a) \leq \max_{\pi \in \Pi} \sum_{s,t \in \pi} \left| \left(B_a^\dagger B_a\right)_{s,t} \right| = \max_{\pi \in \Pi} \sum_{s,t \in \pi} \left| \left(\left(\mathbf{P}_a \mathbf{W}_T\right)^\dagger \mathbf{P}_a \mathbf{W}_T\right)_{s,t} \right|. \tag{12}$$

This sensitivity allows us to compute a $\mu$-GDP bound, which we then convert into RDP. Since the initial accounting in Cyffers et al. (2022) was performed at the user level rather than at the finer-grained record level, we run our accounting under the worst-case assumption that the attacked node always exposes the same single record (this corresponds to $(T, 1)$-participation for all nodes). This ensures that our guarantees are computed at the same user level, leading to a fair comparison.

Note that the accounting does not depend on the choice of learning model or dataset. Rather, it is determined by the participation scheme, the number of epochs, the clipping parameter, and the noise scale, but also on the communication graph through the gossip matrix, as PNDP provides privacy guarantees that are specific to each pair of nodes. We thus consider several graphs that correspond to plausible real-world scenarios:

- *Erdős–Rényi graphs*: generated randomly with $100$ nodes and parameter $p = 0.2$, ensuring that each generated graph is strongly connected. These graphs have expander properties and can be generated in a distributed fashion.
- *Facebook Ego graph (Leskovec & Mcauley, 2012)*: corresponds to the graph of a fixed Facebook user (omitted from the graph), where nodes are the user's friends and edges represent friendships between them. This graph has $148$ nodes.
- *PeerTube graph (Damie & Cyffers, 2026)*: an example of decentralized social platform. PeerTube is an open-source decentralized alternative to YouTube. Each node represents a PeerTube server and edges encode follow relationships between servers, allowing users to watch recommended videos across instances. The graph, restricted to its largest connected component, has $271$ nodes.
- *Florentine Families (Breiger & Pattison, 1986)*: a historical graph with $15$ nodes describing marital relations between families in 15th-century Florence.

Results are shown in Figure 2. For all graphs, our accounting is significantly tighter for all possible distances between nodes. In particular, while the PNDP accounting of Cyffers et al. (2022) provides no improvement over LDP guarantees for nodes at distances less than or equal to $2$, our method already achieves significant gains—up to an order of magnitude. The improvement is even larger, with reductions of at least two orders of magnitude at distances greater than or equal to $3$. The gains are consistent across all the considered graphs, demonstrating the versatility of our method, which leverages both the topology and the correlations induced between nodes.

## 7.2 MAFALDA-SGD

We now illustrate the performance of MAFALDA-SGD under LDP and compare it with three baselines: non-private D-SGD, AntiPGD as defined in Appendix A, and standard DP-D-SGD without noise correlation between nodes. Results on the Facebook Ego graph are shown in Figure 3, with additional results for other privacy levels and graphs reported in Appendix C. We use our accounting for all the algorithms to provide a fair comparison and because it is the tightest accounting available.

We study a regression task on the Housing dataset,[1] which has $8$ features and $20{,}640$ data points, with the goal of predicting house prices based on demographic and housing features. In all experiments, we use a simple multilayer perceptron (MLP) with one hidden layer of width $64$ and ReLU activation, followed by a linear output layer for regression, trained with the mean-square error loss. This dataset has been used in several related works (Cyffers et al., 2022; Li et al., 2025). In all experiments, we cycle over the data points such that each of them participates $20$ times in total, with participation every $19$ steps. These numbers were chosen for memory constraints and not optimized. This cyclic participation corresponds to a $(k, b)$-participation scheme with $k = 20$ and $b = 19$ in the matrix factorization literature (Choquette-Choo et al., 2023).

Our algorithm MAFALDA-SGD outperforms all private baselines across all privacy regimes by a large margin. For a fixed privacy budget $\epsilon$, MAFALDA-SGD achieves a $31\%$ average improvement in test loss in the last $50$ training steps, while for a fixed test-loss value of $0.75$, it achieves a 2-fold reduction in $\epsilon$. These results illustrate the importance of correlated noise for the performance of differentially private decentralized algorithms. Additional experiments confirm that this superiority holds regardless of the graph topology. In particular, under small privacy budgets, there exist regimes where MAFALDA-SGD converges while competitors diverge (e.g., AntiPGD in Figure 3).

---

[1] https://www.openml.org/d/823

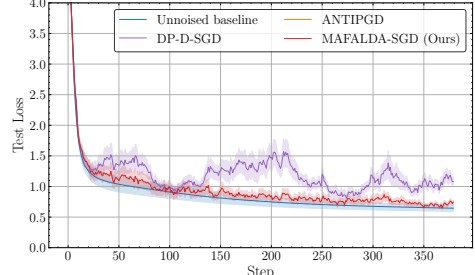 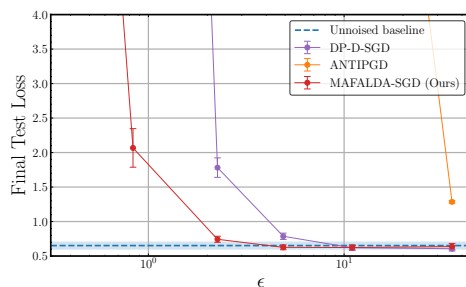

Figure 3: Comparison of MAFALDA-SGD with the non-private baseline, AntiPGD, and standard DP-SGD under local differential privacy on the Housing dataset with the Facebook Ego graph. Left: test loss over time (AntiPGD does not appear because the test loss is always greater than 6), averaged over 20 runs. Right: final test loss as a function of the LDP privacy budget, averaged over 20 runs. In both cases, data points are distributed uniformly at random across nodes.

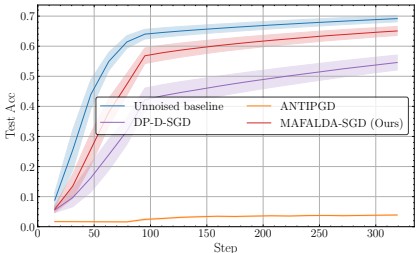 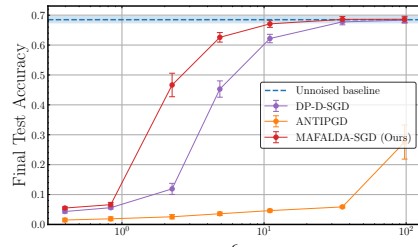

Figure 4: FEMNIST classification results on the Facebook Ego graph. Left: test accuracy evolution for a fixed privacy budget. Right: privacy-utility tradeoff between the final test accuracy as a function of the LDP privacy budget. Error bands show 95% confidence intervals.

We also evaluate MAFALDA-SGD on the federated EMNIST (FEMNIST) image classification task from the LEAF benchmark (Caldas et al., 2018), predicting handwritten characters across users. This setting is highly non-iid and realistic for decentralized learning. To ensure enough data is present, each node is given 10 unique handwriting styles as training data. We use a small convolutional network comprising two convolutional layers (16 and 32 filters, $3 \times 3$ kernels) with ReLU and $2 \times 2$ max-pooling, followed by group normalization and a fully-connected layer for a softmax output trained with cross-entropy. Training uses a cyclic participation scheme $((k, b) = (20, 16))$.

We compare MAFALDA-SGD to baselines under local DP accounting. Results (averaged over 20 runs) are reported in Figure 4. Across all privacy budgets, MAFALDA-SGD consistently achieves higher test accuracy than the private baselines, with the largest relative gains at high privacy levels where correlated noise preserves utility while satisfying LDP.

## 8 CONCLUSION

In this work, we establish a new connection between two separate research directions by extending the matrix factorization mechanism from the well-studied centralized DP-SGD to differentially private decentralized learning. This required generalizing known results in matrix factorization to broader classes of matrices, which may also prove useful in other contexts. Our framework is flexible enough to capture both algorithms and trust models, while providing tighter privacy guarantees than prior analyses. It also enables the design of new algorithms, as illustrated by MAFALDA-SGD, which outperforms existing approaches. Overall, our framework lays the foundation for a more principled design of private decentralized algorithms and enhances the practicality of privacy-preserving machine learning in decentralized settings.

ACKNOWLEDGMENTS

This work was supported by grant ANR-20-CE23-0015 (Project PRIDE) and the ANR 22-PECY-0002 IPOP (Interdisciplinary Project on Privacy) project of the Cybersecurity PEPR and by the Austrian Science Fund (FWF) [10.55776/COE12]. This research was supported in part by the Groupe La Poste, sponsor of the Inria Foundation, in the framework of the FedMalin Inria Challenge. This project was provided with computing AI and storage resources by GENCI at IDRIS thanks to the grant 2025-AD011015352R1 on the supercomputer Jean Zay's V100 partition. Experiments presented in this paper were also carried out using the Grid'5000 testbed, supported by a scientific interest group hosted by Inria and including CNRS, RENATER and several Universities as well as other organizations (see `https://www.grid5000.fr`). The authors thank Quino for his comic strip Mafalda which inspired our new algorithm name.

REPRODUCIBILITY

We provide an open-source implementation of the code used in this work, publicly available at `https://github.com/dimiarbre/MFDL`.

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

| Symbol | Usage | Source |
|---|---|---|
| $\mathcal{V}$ | Set of nodes participating in the training | |
| $n$ | Number of nodes in the training | |
| $d$ | Model size | |
| $\eta$ | Learning rate | |
| $T$ | time $t$ ranges from 1 to $T$ | |
| $\Phi$ | Number of observations made by the attacker | |
| $\theta$ | A model during the training | |
| $W$ | Gossip matrix | |
| $\mathbf{W}_T$ | Decentralized learning workload | Figure 1 |
| $\mathbf{C}^\top$ | Transpose of matrix $\mathbf{C}$. | |
| $\mathbf{C}^\dagger$ | Moore-Penrose pseudo-inverse of matrix $\mathbf{C}$ | |
| $\otimes$ | Kronecker product | |
| $\Pi$ | Participation scheme | |
| $\mathcal{D}_u$ | Dataset of some node $u$ | |

Table 1: List of symbols used in this work

## A    SOTA DP-DL APPROACHES AS MATRIX FACTORIZATION MECHANISMS

In this section, we prove Theorem 6 by looking at algorithms one by one. But first, we explain the strategy used to get the factorization, and we present a useful Lemma.

### A.1    PATH TO MATRIX FACTORIZATION MECHANISMS

For each setting, the proof of Theorem 6 follows a two-step strategy:

1. list observed messages, known gradients, and known noise, and derive the corresponding matrices $A$ and $B$;

2. (if necessary) remove the impact of the known gradients from matrix $A$ (apply Lemma 14).

Apart from LDP, the second step is mandatory to recover a proper factorization. Indeed, if one considers a Linear DL algorithm and an attacker knowledge $\mathcal{O}_\mathcal{A} = AG + BZ$ such that $A = \begin{bmatrix} \bar{A} \\ K^G \end{bmatrix}$ and $B = \begin{bmatrix} \bar{B} \\ 0 \end{bmatrix}$. Then, no matrix $C$ is such that $A = BC$, as it would imply that $K^G = 0C$.

However, by following the cleaning strategy given by Lemma 14, we obtain a second reduced view, $\tilde{\mathcal{O}}_{\mathcal{A}} := \tilde{A}G + BZ$, such that $\tilde{A} = BC$ can exist.

**Lemma 14** (Attacker-knowledge reduction). *Let $\mathcal{O}_{\mathcal{A}} = AG + BZ$ be the attacker knowledge of a trust model on a Linear DL algorithm, such that the attacker is aware of some linear combinations of gradients (without noise). Without loss of generality, let assume $A = [\bar{A}^{\top} \ K^{G\top}]^{\top}$ and $B = [\bar{B}^{\top} \ 0]^{\top}$ where none of $\bar{B}$'s rows are null. If we define $\tilde{A} := A(I - K^{G\dagger}K^G)$, then the GDP guarantees of the attacker knowledge $\mathcal{O}_{\mathcal{A}}$ are the same as the one of the* attacker-knowledge reduction $\tilde{\mathcal{O}}_{\mathcal{A}} := \tilde{A}G + BZ$.

*Proof.*

$$AG + BZ = A\left(K^{G\dagger}K^G\right)G + A\left(I - K^{G\dagger}K^G\right)G + BZ \tag{13}$$

$$= AK^{G\dagger}\left(K^G G\right) + \begin{bmatrix} \bar{A}\left(I - K^{G\dagger}K^G\right) \\ K^G\left(I - K^{G\dagger}K^G\right) \end{bmatrix} G + BZ \tag{14}$$

$$= AK^{G\dagger}\left(K^G G\right) + \begin{bmatrix} \bar{A}\left(I - K^{G\dagger}K^G\right) \\ 0 \end{bmatrix} G + BZ \tag{15}$$

$$= AK^{G\dagger}\left(K^G G\right) + \begin{bmatrix} \bar{A} \\ 0 \end{bmatrix}\left(I - K^{G\dagger}K^G\right) G + BZ \tag{16}$$

$$= AK^{G\dagger}\left(K^G G\right) + \tilde{A}G + BZ \tag{17}$$

$$\tag{18}$$

Then, as the attacker knows $K^G G$, the GDP properties of $AG + BZ$ are the same as those of $\tilde{A}G + BZ$, which concludes the proof.

$\square$

## A.2 Proof of Theorem 6 for DP-D-SGD

In this section, we make explicit the factorization of Theorem 6 for the DP-SGD algorithm, where all noises are independent, under the different trust models.

**LDP** With LDP, the attacker has only access to messages, therefore, no reduction is needed, and we analyze the factorization of the messages directly. We have $C_{\text{DP–SGD}} = I_{nT}$, yielding $A_{\text{DP-D-SGD}} = B_{\text{DP-D-SGD}} = \mathbf{W}_T$.

**PNDP** PNDP considers a set $\mathcal{A}$ of attacker nodes. Each attacker observes

- the messages it receives,
- its own gradients,
- its own noise,
- the messages it emits (which is a linear combination of previous information, so we will omit it).

To ease the proof, without loss of generality, we reorder G and Z such that their first lines correspond to the attackers' gradients and noise.

Define $\mathbf{K}_{\text{PNDP}}^{\mathcal{S}}$ as:

$$\mathbf{K}_{\text{PNDP}}^{\mathcal{S}} = I_T \otimes K_{\mathcal{A}} \tag{19}$$

where $K_{\mathcal{A}}[i, \Gamma_{\mathcal{A}}[i]] = 1$ for $i \in [\![1, |\Gamma_{\mathcal{A}}|]\!]$, where $\Gamma_{\mathcal{A}} = \bigcup_{a \in \mathcal{A}} \Gamma_a$ represents the (ordered) list of neighbors of $\mathcal{A}$. $\mathbf{K}_{\text{PNDP}}^{\mathcal{S}}$ thus projects the set of all messages onto the set of messages received by $\mathcal{A}$. Denote $a$ the number of attackers, and $b$ the total number of their neighbors.

Then, the attacker knowledge is

$$
\begin{bmatrix} (\mathbf{K}_{\text{PNDP}}^{\mathcal{S}} \mathbf{W}_T) \\ I_{aT} \quad 0 \\ 0 \quad 0 \end{bmatrix} G + \begin{bmatrix} (\mathbf{K}_{\text{PNDP}}^{\mathcal{S}} \mathbf{W}_T) \\ 0 \quad 0 \\ I_{aT} \quad 0 \end{bmatrix} Z,
$$

where $Z \in \mathbb{R}^{nT \times d}$. After reduction, it becomes

$$
\begin{bmatrix} (\mathbf{K}_{\text{PNDP}}^{\mathcal{S}} \mathbf{W}_T) \\ 0 \quad 0 \\ 0 \quad 0 \end{bmatrix} \begin{bmatrix} 0 & 0 \\ 0 & I_{(n-a)T} \end{bmatrix} G + \begin{bmatrix} (\mathbf{K}_{\text{PNDP}}^{\mathcal{S}} \mathbf{W}_T) \\ 0 \quad 0 \\ I_{aT} \quad 0 \end{bmatrix} Z.
$$

Hence $A = B \begin{bmatrix} 0_{aT \times aT} & 0_{aT \times (n-a)T} \\ 0_{(n-a)T \times aT} & I_{(n-a)T} \end{bmatrix}$, which concludes the proof for PNDP setting.

## A.3 PROOF OF THEOREM 6 FOR ANTI-PGD

In this section, we make explicit the factorization of Theorem 6 for the AntiPGD algorithm, which was studied in Koloskova et al. (2023).

**LDP**  Following the factorization found in a centralized setting (Koloskova et al., 2023), we have $C_{\text{ANTI-PGD}} = 1_{t \geq t'} \otimes I_n$, $A_{\text{ANTI-PGD}} = \mathbf{W}_T$ and:

$$
B_{\text{ANTI-PGD}} = \begin{bmatrix} I_n & 0 & \dots & 0 \\ W - I_n & I_n & \dots & 0 \\ \vdots & \vdots & \ddots & \vdots \\ W^{T-2}(W - I_n) & W^{T-3}(W - I_n) & \dots & I_n \end{bmatrix}
$$

This notation shows how noise cancellation is propagated and lagging one step behind in terms of communication.

**PNDP**  PNDP considers a set $\mathcal{A}$ of attacker nodes. To ease the proof, without loss of generality, we reorder G and Z such that their first lines correspond to the attackers gradients and noise.

Define $\mathbf{K}_{\text{PNDP}}^{\mathcal{S}}$ as the projection of all messages onto the set of messages received by $\mathcal{A}$. Denote $a$ the number of attackers, and $b$ the total number of their neighbors.

An alternative expression of messages in ANTI-PGD is $\widetilde{\mathbf{W}}_T (G + DZ)$, where $D$ is the invertible

lower-triangular matrix $\begin{bmatrix} 1 & & & \\ -1 & 1 & & 0 \\ 0 & -1 & 1 & \\ \vdots & \ddots & \ddots & \ddots \\ 0 & \dots & 0 & -1 & 1 \end{bmatrix}$.

Then, the attacker knowledge is

$$
\begin{bmatrix} (\mathbf{K}_{\text{PNDP}}^{\mathcal{S}} \mathbf{W}_T) \\ I_{aT} \quad 0 \\ 0 \quad 0 \end{bmatrix} G + \begin{bmatrix} (\mathbf{K}_{\text{PNDP}}^{\mathcal{S}} \mathbf{W}_T D) \\ 0 \quad 0 \\ I_{aT} \quad 0 \end{bmatrix} Z,
$$

where $Z \in \mathbb{R}^{nT \times d}$. After reduction, it becomes

$$
\begin{bmatrix} (\mathbf{K}_{\text{PNDP}}^{\mathcal{S}} \mathbf{W}_T) \\ 0 \quad 0 \\ 0 \quad 0 \end{bmatrix} \begin{bmatrix} 0 & 0 \\ 0 & I_{(n-a)T} \end{bmatrix} G + \begin{bmatrix} (\mathbf{K}_{\text{PNDP}}^{\mathcal{S}} \mathbf{W}_T D) \\ 0 \quad 0 \\ I_{aT} \quad 0 \end{bmatrix} Z.
$$

Hence $A = BD^{-1} \begin{bmatrix} 0_{aT \times aT} & 0_{aT \times (n-a)T} \\ 0_{(n-a)T \times aT} & I_{(n-a)T} \end{bmatrix}$, which concludes the proof for PNDP setting.

## A.4 PROOF OF THEOREM 6 FOR MUFFLIATO-SGD

In this section, we make explicit the factorization of Theorem 6 for the Muffliato algorithm, which was studied in Cyffers et al. (2022).

**LDP**  Muffliato-SGD (Cyffers et al., 2022) requires a different communication workload than $\mathbf{W}_T$. That is because they consider multiple communication rounds before computing a new gradient. Formally, they define $K$ the number of communication steps between two gradients, and they perform $T$ iterations. This means that the set of messages shared is:

$$\mathcal{O}_{\text{Muffliato-SGD}} = \mathbf{W}_{KT} \left( I_T \otimes \begin{bmatrix} I_n \\ \mathbf{0}_{Kn \times n} \end{bmatrix} \right) (G + Z).$$

We see here that Muffliato-SGD adds uncorrelated local noises ($C^\dagger = I_{nT}$), and thus we have $C_{\text{Muffliato-SGD}} = I_{nT}$, as well as $A_{\text{Muffliato-SGD}} = B_{\text{Muffliato-SGD}} = \mathbf{W}_{KT} \left( I_T \otimes \begin{bmatrix} I_n \\ \mathbf{0}_{(K-1)n \times n} \end{bmatrix} \right)$.

**PNDP**  For Muffliato-SGD(K), we assume $T = KT'$, where $T'$ is the number of gradient descent, and has $K$ communication rounds between each gradient descent.

PNDP considers a set $\mathcal{A}$ of attacker nodes. To ease the proof, without loss of generality, we reorder G and Z such that their first lines correspond to the attackers gradients and noise.

Define $\mathbf{K}_{\text{PNDP}}^{\mathcal{S}}$ as the projection of all messages onto the set of messages received by $\mathcal{A}$. Denote $a$ the number of attackers, and $b$ the total number of their neighbors.

Then, the attacker knowledge is

$$\begin{bmatrix} \left(\mathbf{K}_{\text{PNDP}}^{\mathcal{S}} A_{\text{Muffliato-SGD}}\right) \\ I_{aT} \quad 0 \\ 0 \quad 0 \end{bmatrix} G + \begin{bmatrix} \left(\mathbf{K}_{\text{PNDP}}^{\mathcal{S}} A_{\text{Muffliato-SGD}}\right) \\ 0 \quad 0 \\ I_{aT} \quad 0 \end{bmatrix} Z,$$

where $Z \in \mathbb{R}^{nT \times d}$. After reduction, it becomes

$$\begin{bmatrix} \left(\mathbf{K}_{\text{PNDP}}^{\mathcal{S}} A_{\text{Muffliato-SGD}}\right) \\ 0 \quad 0 \\ 0 \quad 0 \end{bmatrix} \begin{bmatrix} 0 & 0 \\ 0 & I_{(n-a)T} \end{bmatrix} G + \begin{bmatrix} \left(\mathbf{K}_{\text{PNDP}}^{\mathcal{S}} A_{\text{Muffliato-SGD}}\right) \\ 0 \quad 0 \\ I_{aT} \quad 0 \end{bmatrix} Z.$$

Hence $A = B \begin{bmatrix} 0_{aT \times aT} & 0_{aT \times (n-a)T} \\ 0_{(n-a)T \times aT} & I_{(n-a)T} \end{bmatrix}$, which concludes the proof for PNDP setting.

Note that while the formulation is similar to the proof for DP-D-SGD, the workload matrix here has $K$ times more rows.

## A.5  Proof of Theorem 6 for DECOR

In this section, we make explicit the factorization of Theorem 6 for the DECOR algorithm, which was studied in Allouah et al. (2024).

First, let us provide the correlation matrix used in DECOR $C_{\text{DECOR}}$. Correlation is performed *edge-wise* in the graph, as each neighbor adds a noise correlated with their neighbors to their local noise. Consider an arbitrary orientation $E$ of edges $\mathcal{E}$ of the symmetric graph $\mathcal{G}$. The correlation matrix has the following structure:

$$C_{\text{DECOR}} = I_T \otimes \begin{bmatrix} I_n & C_{\text{nodes}}^\dagger \end{bmatrix}^\dagger$$

$$\forall e \in [\![1, |E|]\!], \forall i \in [\![1, n]\!] : C_{\text{nodes}}^\dagger[i, e] = \begin{cases} 1 \text{ if } E[e] = (i, j) \\ -1 \text{ if } E[e] = (j, i) \\ 0 \text{ otherwise} \end{cases}.$$

where the first part of $\begin{bmatrix} I_{nT} & C^\dagger \end{bmatrix}$ corresponds to the local private noise, and the second part to the "secrets". Interestingly, multiple correlation matrices (and thus equivalent factorizations) exist for DECOR, as there are multiple ways to generate possible orientations of the edges of the graph $\mathcal{G}$.

**LDP**  The messages exchanged by DECOR take the form $-\eta \widetilde{\mathbf{W}}_T \left( G + C_{\text{DECOR}}^\dagger Z \right)$. Hence $A = \widetilde{\mathbf{W}}_T$, $B = \widetilde{\mathbf{W}}_T \begin{bmatrix} I_{nT} & C^\dagger \end{bmatrix}$, and $A = B \begin{bmatrix} I_{nT} \\ 0 \end{bmatrix}$, which concludes the proof for LDP setting.

**PNDP**  With DECOR, each node knows some of the noises of its neighbors. Therefore, Sec-LDP is more appropriate to describe the knowledge gathered by some attackers.

**Sec-LDP**  Let us consider a set $\mathcal{A}$ of attacker nodes. To ease the proof, without loss of generality, we reorder G such that its first lines correspond to the attackers gradients. For Z, we first gather the local noise of the attackers, then other local noises, and finally the "secrets" noise.

Define $\mathbf{K}_{\text{PNDP}}^{\mathcal{S}}$ as the projection of all messages onto the set of messages received by $\mathcal{A}$. Denote $a$ the number of attackers.

Then, the attacker knowledge is

$$
\begin{bmatrix} \mathbf{K}_{\text{PNDP}}^{\mathcal{S}}\mathbf{W}_T \\ I_{aT} \quad 0 \\ 0 \quad 0 \\ 0 \quad 0 \end{bmatrix} G + \begin{bmatrix} \mathbf{K}_{\text{PNDP}}^{\mathcal{S}}\widetilde{\mathbf{W}}_T \begin{bmatrix} I_{nT} & D^\dagger \end{bmatrix} \\ 0 \quad 0 \quad 0 \\ I_{aT} \quad 0 \quad 0 \\ 0 \quad 0 \quad E \end{bmatrix} Z,
$$

where $Z \in \mathbb{R}^{nT \times d}$ and $D$ is the correlation matrix of secrets and $D$ a subpart of $E$. After reduction, it becomes

$$
\begin{bmatrix} \mathbf{K}_{\text{PNDP}}^{\mathcal{S}}\widetilde{\mathbf{W}}_T \\ 0 \quad 0 \\ 0 \quad 0 \\ 0 \quad 0 \end{bmatrix} \begin{bmatrix} 0 & 0 \\ 0 & I_{(n-a)T} \end{bmatrix} G + \begin{bmatrix} \mathbf{K}_{\text{PNDP}}^{\mathcal{S}}\widetilde{\mathbf{W}}_T \begin{bmatrix} I_{nT} & D^\dagger \end{bmatrix} \\ 0 \quad 0 \quad 0 \\ I_{aT} \quad 0 \quad 0 \\ 0 \quad 0 \quad E \end{bmatrix} Z.
$$

Hence $A = B \begin{bmatrix} 0_{aT \times aT} & 0_{aT \times (n-a)T} \\ 0_{(n-a)T \times aT} & I_{(n-a)T} \\ 0_{b \times aT} & 0_{b \times (n-a)T} \end{bmatrix}$, where $b$ is the number of columns of $D^\dagger$, which concludes the proof for PNDP setting.

Note that while the formulation is similar to the proof for DP-D-SGD, the workload matrix here has $K$ times more rows.

### A.6  PROOF OF THEOREM 6 FOR ZIP-DL

In this section, we make explicit the factorization of Theorem 6 for the Zip-DL algorithm, which was introduced in Biswas et al. (2025).

Zip-DL considers message-wise noises, with one noise per message sent between nodes. For each node $i$, let $\Gamma_i$ be its set of neighbors. Define $s_i = |\Gamma_i|$ the number of messages sent by $i$ and $s = \sum_{i=1}^{n} s_i$ the total number of messages.

Zip-DL is already expressed under a matrix form (Biswas et al., 2025): by introducing $n^2$ virtual nodes, and translating to express the gossip matrix $W$ under this virtual equivalent as $\widetilde{W} \in \mathbb{R}^{n^2 \times n^2}$ to encode messages. Formally, we have $\widetilde{W}[ni+j+1, nj+i+1] = W[i,j]$. Furthermore, considering an averaging matrix $\widetilde{M}$, and a correlation matrix $C_{\text{Zip-DL}}^\dagger$ (coined $\hat{C}$ in Biswas et al. (2025)), Zip-DL can be expressed as:

$$\theta_{t+1} = \widetilde{M}\widetilde{W} \left( 1_n(\theta_t - \eta G_t) + C_{\text{Zip-DL}}^\dagger Z \right) \tag{20}$$

We can now define the messages shared on the network, and viewed by an LDP attacker $\mathcal{A}$:

$$\mathcal{O}_\mathcal{A} = \widetilde{\mathbf{W}}_T \left( (I_{nT} \otimes 1_n) G + \left( I_T \otimes C_{\text{Zip-DL}}^\dagger \right) Z \right), \tag{21}$$

$$\widetilde{\mathbf{W}}_T = \left( I_T \otimes \tilde{I}_{n^2} \right) \begin{bmatrix} I_{n^2} & 0 & \dots & 0 \\ \widetilde{M}\widetilde{W} & I_{n^2} & \dots & 0 \\ \vdots & \vdots & \ddots & \vdots \\ \left(\widetilde{M}\widetilde{W}\right)^{T-1} & \left(\widetilde{M}\widetilde{W}\right)^{T-2} & \dots & I_{n^2} \end{bmatrix} \tag{22}$$

where $\tilde{I}_{n^2}$ is a diagonal matrix filled with zeros except for $\tilde{I}_{n^2}[ni+j+1, ni+j+1] = 1 \iff j \in \Gamma_i$. Intuitively, $\widetilde{M}\widetilde{W}$ corresponds to one round of averaging. This is typically a very sparse matrix, even more so because many lines will be zero when considering communication graphs that are not the complete graph.

**LDP** $A = \widetilde{\mathbf{W}}_T (I_{nT} \otimes 1_n)$, $B = \widetilde{\mathbf{W}}_T (I_{nT} \otimes 1_n) \left( I_T \otimes C_{\text{Zip-DL}}^{\dagger} \right)$, and $C = (I_T \otimes C_{\text{Zip-DL}})$ concludes the proof for Zip-DL algorithm.

**PNDP** For the ease of notation, we assume here that for any node $i$, $s_i = c$. The generalization is straightforward. To ease the proof, without loss of generality, we reorder $G$ and $Z$ such that their first lines correspond to the attackers gradients and noise.

We denote $\mathcal{A}$ the set of attacker nodes, and assume for simplicity there is only one such attacker node (extension to more simply requires doing this decomposition multiple times, or extending the projection matrices). Denote $\mathbf{K}_{\text{PNDP}}^{\mathcal{S}} \in \mathbb{R}^{cT \times cnT}$ the matrix selecting the messages received by $\mathcal{A}$.

The PNDP attacker knowledge is therefore

$$
\begin{bmatrix}
\left( \mathbf{K}_{\text{PNDP}}^{\mathcal{S}} \widetilde{\mathbf{W}}_T \right) & \\
(I_T \otimes [1 \quad \mathbf{0}_{1 \times c-1}]) & 0 \\
0 & 0
\end{bmatrix}
(I_{nT} \otimes \mathbf{1}_c) G +
\begin{bmatrix}
\left( \mathbf{K}_{\text{PNDP}}^{\mathcal{S}} \widetilde{\mathbf{W}}_T \right) & \\
0 & 0 \\
I_{cT} & 0
\end{bmatrix}
Z,
$$

where $Z \in \mathbb{R}^{ncT \times d}$. After reduction, it becomes

$$
\begin{bmatrix}
\left( \mathbf{K}_{\text{PNDP}}^{\mathcal{S}} \widetilde{\mathbf{W}}_T \right) \\
0 \quad 0 \\
0 \quad 0
\end{bmatrix}
\begin{bmatrix}
0 & 0 \\
0 & \left( I_{(n-1)T} \otimes \mathbf{1}_c \right)
\end{bmatrix}
G +
\begin{bmatrix}
\left( \mathbf{K}_{\text{PNDP}}^{\mathcal{S}} \widetilde{\mathbf{W}}_T \right) \\
0 \quad 0 \\
I_{cT} \quad 0
\end{bmatrix}
Z.
$$

Hence $A = B \begin{bmatrix} 0_{cT \times cT} & 0_{cT \times (n-1)cT} \\ 0_{(n-1)cT \times cT} & I_{(n-1)T} \otimes \mathbf{1}_c \end{bmatrix}$, which concludes the proof for Zip-DL algorithm.

## A.7 EXTENDING NOTATIONS TO TIME-VARYING GOSSIP

We extend the notations introduced in Section 4.1 to time-varying graphs encoded as a sequence of gossip matrices $W_t$, for $t \in [\![1, T]\!]$. In this more general context, the workload matrix $\mathbf{W}_T$ becomes:

$$
\mathbf{W}_T =
\begin{bmatrix}
I_n & 0 & 0 & \cdots & 0 \\
W_1 & I_n & 0 & \cdots & 0 \\
W_2 W_1 & W_2 & I_n & \cdots & 0 \\
\vdots & \vdots & \vdots & \ddots & \vdots \\
\Pi_{t=T}^1 W_t & \Pi_{t=T-1}^2 W_t & \Pi_{t=T-1}^3 W_t & \cdots & I_n
\end{bmatrix}
$$

This new definition of $\mathbf{W}_T$ encodes how messages spread through time using the sequence of gossip matrices $(W_t)_t$. The formalization of the MF-D-SGD algorithm (initially presented in Equation (5)) similarly needs adapting by replacing $I_T \otimes W$ by a block diagonal matrix of all $W_t$:

$$
\theta =
\begin{bmatrix}
W_1 & 0 & \cdots & 0 \\
0 & W_2 & \cdots & 0 \\
\vdots & \vdots & \ddots & \vdots \\
0 & 0 & \cdots & W_T
\end{bmatrix}
\left( M\theta_0 - \eta \, \mathbf{W}_T \left( G + C^{\dagger} Z \right) \right),
\tag{23}
$$

## B PROOF OF THEOREM 8

*Proof.* (of Theorem 8)

We denote $\mathcal{M} \equiv \mathcal{M}'$ two mechanisms that have the same $\mu$-GDP properties. In this proof, this will either be because they are distributionally equivalent, or because they are an invertible operation away from each other (meaning they have the same $\mu$-GDP properties by post-processing).

We start by proving the following equivalences:

$$\mathcal{M} : G \to B(CG + Z)$$
$$\equiv \mathcal{M}_1 : G \to B'(CG + Z) \qquad \triangleright \text{ with } B' \in \mathbb{R}^{\text{rank}(B) \times m} \text{ and } B'C \text{ an echelon matrix}$$
$$\equiv \mathcal{M}_2 : G \to [L\,0]\,(QCG + Z) \qquad \triangleright \text{ with } B' = [L\,0]\,Q \text{ the LQ decomposition of } B'$$
$$\equiv \mathcal{M}_3 : G \to (\tilde{Q}CG + \tilde{Z}) \qquad \triangleright \text{ with } \tilde{Q} \text{ (resp. } \tilde{Z}) \text{ the first rank } (B') \text{ rows of } Q \text{ (resp. } Z)$$

Then, since $\mathcal{M}_3$ can operate in the continual release model, those privacy guarantees will transfer back to $\mathcal{M}$.

---

**Algorithm 3** Rows Selection for Matrix $B$

---

**Require:** Matrix $B \in \mathbb{R}^{a \times b}$
1:  $R \leftarrow []$                                                $\triangleright$ Selected rows (empty matrix)
2:  $P \leftarrow []$
3: **for** $i = 1$ to $a$ **do**
4:     **if** $B_{[i]}$ is not a linear combination of rows in $R$ **then**
5:         $R \leftarrow \begin{pmatrix} R \\ B_{[i]} \end{pmatrix}$
6:         $P \leftarrow \begin{pmatrix} P \\ e_i^\top \end{pmatrix}$                  $\triangleright e_i \in \mathbb{R}^a$ selects line $i$ of BC
7: **return** $P$

---

**Proof that $\mathcal{M} \equiv \mathcal{M}_1$:** Let $P$ be a projection matrix selecting $\text{rank}(B)$ rows of $B$, in the original order, and such that $\text{rank}(PB) = \text{rank}(B)$. $P$ can be obtained through Algorithm 3. Denote $B'$ the matrix $PB$. By design, $B'$ is full-rank, $B' \in \mathbb{R}^{\text{rank}(B) \times m}$, $\text{rank}(B') = \text{rank}(B) \leqslant m$, and $B'C$ is an echelon matrix.

Denote $S$ the rows of $B$ not in $B'$. By design, they are linear combinations of rows of $B'$, meaning there exists a matrix $D$ such that $S = DB'$.

As $S$ is the complement of $B'$, there exists a permutation matrix $\pi$ such that

$$B = \pi \begin{bmatrix} B' \\ S \end{bmatrix} = \pi \begin{bmatrix} B' \\ DB' \end{bmatrix} = \pi \begin{bmatrix} I_{\text{rank } B} \\ D \end{bmatrix} B',$$

and finally $\mathcal{M} = \pi \begin{bmatrix} I_{\text{rank } B} \\ D \end{bmatrix} \mathcal{M}_1$.

As also $\mathcal{M}_1 = P\mathcal{M}$, by applying post-processing in both directions, we have $\mathcal{M} \equiv \mathcal{M}_1$.

**Proof of $\mathcal{M}_1 \equiv \mathcal{M}_2$ :** Consider an $LQ$-decomposition of $B'$ : since $\text{rank}(B') = \text{rank}(B) \leq m$ we can find a decomposition $B' = [L \quad 0]\,Q$ such that $L \in \mathbb{R}^{\text{rank}(B') \times \text{rank}(B')}$ is lower triangular and $Q \in \mathbb{R}^{m \times m}$ is an orthonormal matrix Trefethen (1997). Then, $\mathcal{M}_1(G) = [L \quad 0]\,(QCG + QZ)$ for any $G$. Since $Q$ is an orthonormal matrix, this is distributionally equivalent to $\mathcal{M}_2$.

**Proof of $\mathcal{M}_2 \equiv \mathcal{M}_3$:** $L$ in $\mathcal{M}_2$ has been constructed to be square and lower triangular matrix of full rank (as $B'$ is of full rank). Thus, $L$ is invertible. This means, by post-processing using an invertible operator $L$, that the privacy guarantees of $\mathcal{M}_2$ are the one of $\begin{bmatrix} I_{\text{rank}(B')} & 0 \end{bmatrix} (QCG + Z)$. It suffices to name $\tilde{Q} = \begin{bmatrix} I_{\text{rank}(B')} & 0 \end{bmatrix} Q$ and $\tilde{Z} = \begin{bmatrix} I_{\text{rank}(B')} & 0 \end{bmatrix} Z$ to see this corresponds to $\mathcal{M}_3$.

**Proof of $\frac{1}{\sigma}$-GDP of $\mathcal{M}_3$:** $\mathcal{M}_3$ is an instance of the Gaussian mechanism on $\tilde{Q}CG$. Since $\tilde{Z} \sim \mathcal{N}\left(0, \nu^2\right)^{\text{rank}(B') \times d}$, this mechanism is $\frac{1}{\sigma}$-GDP if $\nu = \sigma \, \text{sens}_\Pi(\tilde{Q}C)$. Additionally, we have $PA = B'C = L\tilde{Q}C$. Then, $\tilde{Q}C = L^{-1}PA$, which implies that $\tilde{Q}C$ is lower echelon (by design of $P$ and $L$). Therefore, for the same value of $\nu$, $\mathcal{M}_3$ remains $\frac{1}{\sigma}$-GDP when considering adaptive gradients.

Note that

$$\mathrm{sens}_{\Pi}(\tilde{Q}C) = \max_{G \simeq_\Pi G'} \left\| \tilde{Q}C\,(G - G') \right\|_F \tag{24}$$

$$= \max_{G \simeq_\Pi G'} \sqrt{\mathrm{tr}\left( (C\,(G - G'))^\top \tilde{Q}^\top \tilde{Q} C\,(G - G') \right)} \tag{25}$$

$$= \max_{G \simeq_\Pi G'} \sqrt{\mathrm{tr}\left( (C\,(G - G'))^\top B^\dagger B C\,(G - G') \right)} \tag{26}$$

$$= \max_{G \simeq_\Pi G'} \| C\,(G - G') \|_{B^\dagger B} \tag{27}$$

$$= \mathrm{sens}_{\Pi}(C; B), \tag{28}$$

where Equation (26) stands as the rows of $\tilde{Q}$ and $B$ span the same space.

Then, applying Lemma 3.15 of Pillutla et al. (2025) allows us to derive the following bound:

$$\mathrm{sens}_{\Pi}(C; B) = \mathrm{sens}_{\Pi}(\tilde{Q}C) \leq \max_{\pi \in \Pi} \sum_{s,t \in \pi} \left| \left( \left(\tilde{Q}C\right)^\top \tilde{Q}C \right)_{s,t} \right| \tag{29}$$

$$\leq \max_{\pi \in \Pi} \sum_{s,t \in \pi} \left| \left( C^\top \tilde{Q}^\top \tilde{Q} C \right)_{s,t} \right| \tag{30}$$

$$\leq \max_{\pi \in \Pi} \sum_{s,t \in \pi} \left| \left( C^\top B'^\dagger B' C \right)_{s,t} \right| \tag{31}$$

$$\leq \max_{\pi \in \Pi} \sum_{s,t \in \pi} \left| \left( C^\top B^\dagger B C \right)_{s,t} \right|, \tag{32}$$

where Equation (29) corresponds to Part (a) of Lemma 3.15 in Pillutla et al. (2025) (generalization to batch releases). We also have equality when the $\left( \left(\tilde{Q}C\right)^\top \tilde{Q}C \right)[t,\tau] > 0$ for all $t, \tau \in \pi$, for all $\pi \in \Pi$).

$\square$

## C  ADDITIONAL EXPERIMENTS

In this section, we provide missing experimental details and results from Section 7.

### C.1  HOUSING PRIVACY-UTILITY TRADEOFF

We simulate all baselines on the Housing dataset for various values of $\sigma$. We display their utility privacy tradeoff, by considering the utility as a function of $\epsilon$ for an $(\epsilon, 1e^{-6})$-DP privacy guarantee. Figure 5 reports the resulting privacy budget. For all graphs, the privacy-utility tradeoff offered by MAFALDA-SGD improves upon that of the existing literature.

Smaller graphs only yield better privacy/utility tradeoffs because data is partitioned into bigger batches on smaller graphs, as there is more data for each node, netting better privacy properties.

### C.2  FEMNIST PRIVACY-UTILITY TRADEOFF

We also include Figure 6 to present the privacy-utility tradeoffs for the FEMNIST dataset on the Florentine families graph and the Facebook Ego graph. In both cases, better test accuracy is consistently reached by MAFALDA-SGD for different values of $\epsilon$.

## D  PROOF OF LEMMA 12

In this section we present the proof of Lemma 12, which is used in Section 6 to justify Algorithm 2.

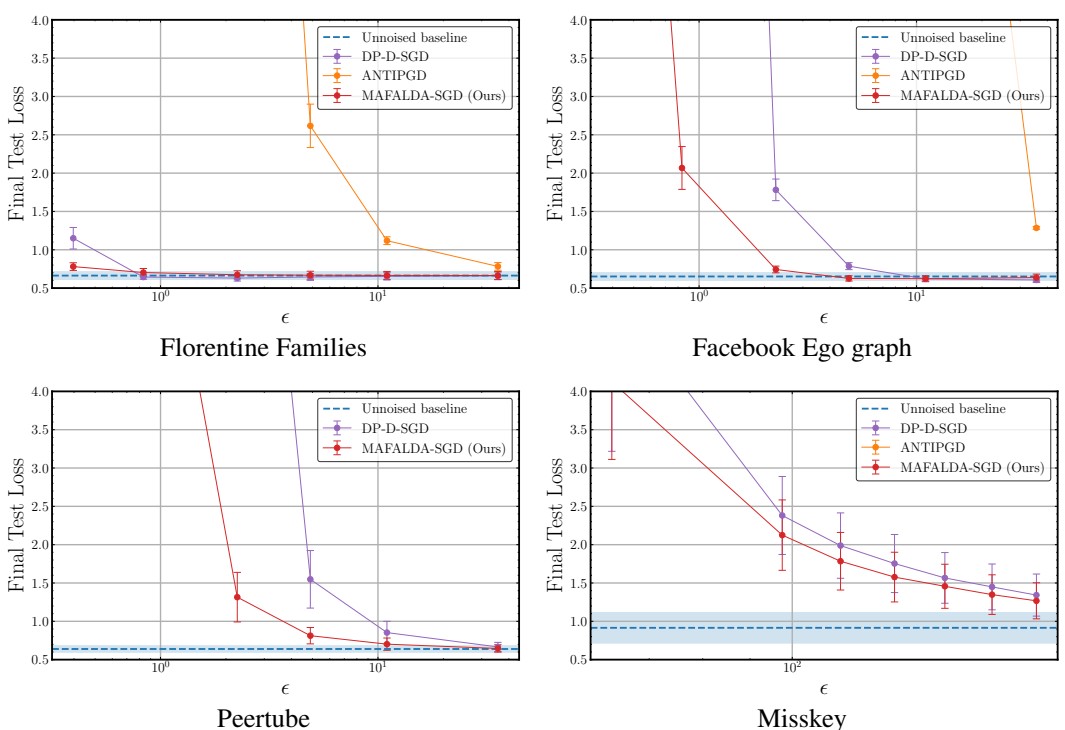

Figure 5: Privacy-utility tradeoff on the housing dataset for various graphs.

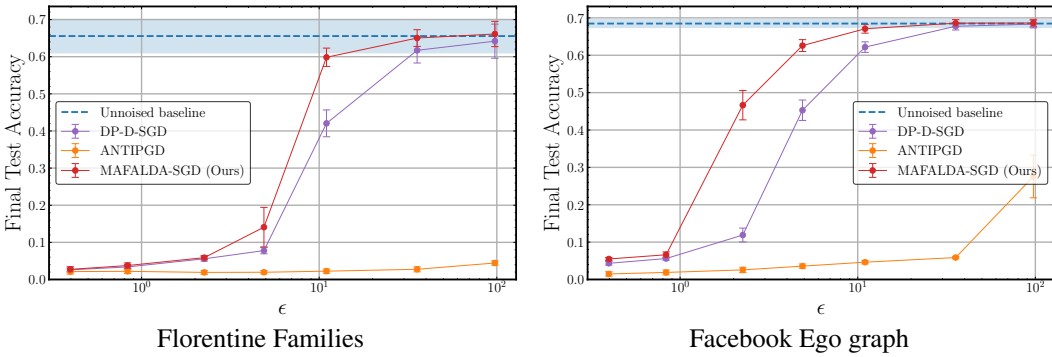

Figure 6: Privacy-utility tradeoff on the FEMNIST dataset for various graphs.

*Proof.* (Lemma 12)

Consider local correlation and $\mathbf{P}_{\mathcal{A}} = I_{nT}$. Then, the optimization loss of Equation (9) becomes:

$$\mathcal{L}_{opti}\left(\mathbf{W}_T, C_{local}\right) = \underset{\Pi_{\text{local}}}{\text{sens}}(C_{local})^2 \left\|\left(I_T \otimes W\right)\mathbf{W}_T\left(C_{local}^{\dagger} \otimes I_n\right)\right\|_F^2 \tag{33}$$

This follows from the property of the Moore-Penrose pseudoinverse: $(C_{local} \otimes I_n)^{\dagger} = C_{local}^{\dagger} \otimes I_n$, as well as the fact that $\text{sens}_{\Pi}(C_{\text{local}} \otimes I_{nT}; B) = \text{sens}_{\Pi_{\text{local}}}(C_{\text{local}})$ under a localized participation scheme $\Pi$ that is identical for all nodes.

As highlighted in Loan (2000), the commutation matrix $\mathbf{K}^{(n,T)}$ (also called the vec-permutation matrix Henderson & Searle (1981)) is a way to change the representation from stacking through time the observations of the global system to stacking the observations locally of each node through time. In other words, for any matrix $X \in \mathbb{R}^{n \times n}, Y \in \mathbb{R}^{T \times T}$, we have:

$$\mathbf{K}^{(n,T)}\left(X \otimes Y\right) = \left(Y \otimes X\right)\mathbf{K}^{(T,n)}$$

It suffices to focus on the norm, since the sensitivity term remains unchanged. Let $A = (I_T \otimes W)\mathbf{W}_T\mathbf{K}^{(n,T)}$. First, note that $\mathbf{K}^{(T,n)}$ is a permutation matrix, so we have:

$$\left\|\left(I_T \otimes W\right)\mathbf{W}_T\left(C_{local}^{\dagger} \otimes I_n\right)\right\|_F^2 = \left\|\left(I_T \otimes W\right)\mathbf{W}_T\left(C_{local}^{\dagger} \otimes I_n\right)\mathbf{K}^{(T,n)}\right\|_F^2$$

$$= \left\|\left(I_T \otimes W\right)\mathbf{W}_T\mathbf{K}^{(n,T)}\left(I_n \otimes C_{local}^{\dagger}\right)\right\|_F^2$$

$$= \left\|A\left(I_n \otimes C_{local}^{\dagger}\right)\right\|_F^2$$

$$= \text{tr}\left(\left[A\left(I_n \otimes C_{local}\right)\right]^{\top}A\left(I_n \otimes C_{local}\right)\right)$$

If we consider $A_i = A_{[:,Ti:T(i+1)-1]}$, then $A\left[I_n \otimes C_{local}\right] = \begin{bmatrix} A_1 C_{local} & \dots & A_n C_{local} \end{bmatrix}$. Thus, when considering the trace, the product will only focus on the blocks for a given node $i$, and there are no cross-terms:

$$\left\|\left(I_T \otimes W\right)\mathbf{W}_T\left(C_{local}^{\dagger} \otimes I_n\right)\right\|_F^2 = \text{tr}\left(\sum_{i=1}^{n}\left(A_i C_{local}\right)^{\top}A_i C_{local}\right)$$

$$= \text{tr}\left(\sum_{i=1}^{n}C_{local}^{\top}A_i^{\top}A_i C_{local}\right)$$

$$= \text{tr}\left(C_{local}^{\top}\left(\sum_{i=1}^{n}A_i^{\top}A_i\right)C_{local}\right).$$

Taking $L$ such that $L^{\top}L = \sum_{i=1}^{n}A_i^{\top}A_i$ using the Cholesky decomposition yields the desired result. $\square$

