# OpenReview forum: "Unified Privacy Guarantees for Decentralized Learning via Matrix Factorization"
_ICLR.cc/2026/Conference — ICLR 2026 Poster_

### Official Review · Reviewer_KGVY · 2025-10-30

**Soundness:** 3
**Presentation:** 2
**Contribution:** 3
**Rating:** 6
**Confidence:** 2

**Summary:**

The paper studies Differentially Private Decentralized Learning and proposes a matrix-factorization view that unifies attacker observations and common trust models (LDP, PNDP, SecLDP). It generalizes prior MF privacy results to a broader class of workloads while preserving adaptivity, enabling tighter privacy accounting for existing DP-DL methods. The framework also disentangles the matrix governing privacy from the one driving optimization, clarifying how to design useful noise correlations in decentralized settings. Building on this, the authors introduce MAFALDA-SGD, which optimizes node-local noise correlations under LDP, and they report tighter PNDP accounting and improved utility on synthetic and real graphs and a regression task.

**Strengths:**

- The work offers a unifying template to express the attacker across multiple decentralized trust models, which generalizes beyond prior algorithm-specific analyses.

- The theoretical contribution extends matrix-factorization privacy guarantees to column-echelon workloads, preserves adaptivity, and formalizes sensitivity, providing principled guarantees that support the framework.

- The empirical results indicate tighter PNDP accounting and improved utility for MAFALDA-SGD under LDP on the reported setups, suggesting practical benefits.

**Weaknesses:**

- The paper’s accessibility is limited for readers without prior exposure to matrix-factorization in DP; the notation is dense, the definition and role of “column-echelon” are introduced with minimal intuition in the main text, and the transition from the general theorem to MAFALDA is abrupt. A short primer with a toy $T=3$ derivation (showing how $A = B C$ and $C^\dagger$ shape correlations), a step-by-step “apply-the-framework” recipe (choose trust model and participation, form the attacker view and objective, impose constraints, solve for correlations), a brief glossary of symbols, and one worked example mapping a familiar DP-D-SGD variant into the proposed matrices would substantially improve readability.

- The empirical scope is narrow, focusing on one regression task and a few graphs; adding a classification task and ablations over participation schemes and correlation patterns would better demonstrate robustness.

- The presentation would benefit from precise definitions and cleanup, including defining the weighted/semi-norm $|\cdot|_{B^\dagger B}$ when first introduced and making the Cholesky orientation consistent throughout (for example, state clearly whether $H = L L^\top$ or $H = L^\top L$ and keep that convention)

- The set of LDP baselines could be expanded or justified; in particular, including DECOR as a comparison (or explaining why it is out of scope) would help situate MAFALDA-SGD among methods that also exploit correlations under LDP.

- Typo: “Independent” in Figure 1

**Questions:**

- Could the authors clarify whether attackers who can read some local models $\theta$ (for example, compromised nodes) can be incorporated by augmenting $O_A = A G + B Z$, and whether the column-echelon and adaptivity arguments still apply in that setting?

- Can you bound the limits of generality by stating which settings fall outside the framework, for example, non-linear message transforms, asynchronous or stale models, quantization or compression, or additional side information that might violate linearity or the column-echelon assumption?

- Do you plan to add further experiments, such as a classification task, ablations over participation schemes $(k,b)$ and training epochs, and sensitivity to graph topology, to substantiate the generality and robustness of the reported improvements?

- Do the authors plan to include an LDP baseline, such as DECOR (or provide a rationale if not), to clarify the relative benefits of optimized local correlations under LDP?

---

> ### Author Response · Authors · 2025-11-20
>
> We thank you for your feedback on presentation and notations. We have uploaded a revised version of the paper with changes highlighted in red. We have performed the following changes to address your remarks:
> 1) We made the Cholesky orientation consistent by defining it in Lemma 11 (now Lemma 12).
> 2) We now define the weigthed norm in Definition 6 (now Definition 7).
>
>
> Now, to address the questions mentionned in the review:
>
> Q1. **Prior knowledge of local models**: Yes, attackers that posses prior knowledge about local models can be incorporated into our setting. For instance, this is the case for a PNDP attacker. Due to space limitations, this encoding was deferred to Appendix A. The column echelon and adaptivity argument still applies, as the observation will follow the causality ordering of DL: only gradients from previous iterations are used at any point. Thus, there exists a representation in column-echelon form even in these scenarios. We extended the paragraph below Theorem 7 (now Theorem 8) to reflect this.
>
> Q2. **Settings falling outside the framework**: Non-linear operations are equivalent to a post processing of the full observations. Thus, the privacy bounds derived in this work still hold. Our theory would, however, not account for the possible privacy gains of such non-linear transforms. Moreover, the notion of optimality used by MAFALDA-SGD will be changed by this non-linearity, meaning that the factorization should be adapted to such scenarios.
>
> Q3. **Further experiments**:  We added an image classification task on the FEMNIST dataset using a CNN in Section 7. We do not plan an ablation study over participation schemes as such patterns are not a novelty of our work, and can be studied in Centralized or Federated settings. We also added a graph topology with more nodes in Figure 4 (now Figure 5) (Misskey graph, 917 nodes) to further study the sensitivity to the graph topology.
>
> Q4. **LDP baseline**: DECOR adds two types of noises to a local model: CDP noises, known only by the node, and Correlated noises between two nodes, using a secret between each pair of nodes. Under LDP, DECOR assumes that an attacker has access to all pairwise correlations that are happening between nodes. Thus, if one were to run an hyperparameter search for the best $(\sigma_{\text{cor}},\sigma_{\text{cdp}})$ under a fixed privacy budget for LDP as performed in the original work, they would naturally find $\sigma_{\text{cor}}=0$, e.g. no correlation between nodes, corresponding to the DP-D-SGD baseline.

---

### Official Review · Reviewer_fuoh · 2025-10-31

**Soundness:** 3
**Presentation:** 3
**Contribution:** 3
**Rating:** 6
**Confidence:** 3

**Summary:**

The paper proposes a unified privacy framework for decentralized learning based on the Matrix Factorization (MF) mechanism, bridging recent advances in centralized DP accounting to decentralized, peer-to-peer training. This paper presents a new algorithm, MAFALDA-SGD, which achieves tighter privacy guarantees.

**Strengths:**

1. The background information and technical introduction are very detailed and the writing is well-organized.
2. The issue of distributed privacy protection is of great practical significance, and the utility-privacy trade-off faced by existing methods is more severe compared to that in the centralized approach.
3. Sufficient theoretical evidence shows that each step has detailed proofs.

**Weaknesses:**

1. The gap between theory and method implementation: The theoretical framework in Sections 4–5 establishes a very general condition for the MF mechanism. This suggests wide applicability to diverse decentralized algorithms and trust models. However, the proposed algorithm MAFALDA-SGD in Section 6 only instantiates this framework under a restricted setting: Local Differential Privacy (LDP) combined with node-local temporal correlation.
2. Additional computational overhead: MAFALDA-SGD algorithm requires an offline optimization of the noise correlation matrix. This process involves computing the Gram matrix and its Cholesky factorization, which may introduce non-negligible computational overhead.
3.	Limited empirical diversity: Experiments focus on small to mid-scale graphs (≤ 300 nodes). Realistic, large-scale federated networks (e.g., > 10⁴ users) would better demonstrate practicality.
4.	Minor clarity issues: Heavy notation could be simplified, and intuitive explanations (especially for Eq. 9 and Lemma 10) would help non-specialists.
5.	Evaluation scope: Only MLP on Housing dataset — a deeper model (e.g., CNN or Transformer under DL setting) would show robustness of MAFALDA-SGD.

**Questions:**

1. Why is a direct multiplicative objective preferable in Eq. 9? Could the authors provide empirical evidence or theoretical justification that this formulation does not suffer from scale imbalance in larger graphs or longer training horizons?
2. Why does the method not explore cross-node correlation or richer participation structures, which the theory explicitly allows?
3. Does the method remain computationally practical when gradients are extremely high-dimensional, as in Transformers?
4. Can your generalized privacy theorem (Theorem 7) apply to asynchronous or time-varying gossip matrices W_t?
5. Did you explore non-Gaussian noise distributions (e.g., Laplace or sub-Gaussian) within the MF framework?
6. Could MAFALDA-SGD be extended to heterogeneous privacy budgets across nodes?
7. Is the matrix-factorization optimization offline-only, or could it adapt online to changing participation patterns?

---

> ### Author Response · Authors · 2025-11-20
>
> We thank the reviewer for their careful review. We have uploaded a revised version of the paper with changes highlighted in red and address the questions they raised below.
>
> Q1. **Scale imbalance**: MAFALDA-SGD optimizes Equation (11), which is a rewriting of Equation (9). It may indeed suffer from scale imbalance, but we avoid this risk as in existing litterature (Choquette-Choo et al, 2023) by constraining $\text{sens}_\Pi(C)$ to be equal to 1.
>
> Q2. **Cross-node correlation**: Our key contribution is Theorem 7 (now Theorem 8), which enables the analysis of DP guarantees for matrix factorizations in DL under diverse participation structures and trust models. Mafalda-SGD illustrates how Theorem 7 (now Theorem 8) can be used in practice in the specific case of LDP and single-node temporal correlation. Cross-node correlation is a natural direction for future research, but raises multiple challenges in terms of computation and privacy-model design. These challenges deserve their dedicated work.
>
> Q3. **High dimensional gradients**: The proposed method remains practical for high-dimensional models: the noise correlation is the same for all parameters. Thus, the correlation matrices do not depend on the number of parameters $d$. Banded constraint (Kalinin & Lampert, 2024) and restarting the correlation (Pillutla et al., 2025) can limit the correlation horizon, reducing the number of noises to store in order to perform the actual correlation. We refer to Remark 13 (new) for details on the computational complexity of our approach.
>
> Q4. **Time-varying matrices**: Time-varying matrices can be expressed in our framework, we detail this in the global rebuttal. Asynchronous gossip matrices can also be encoded in a similar manner, and Theorem 7 (now Theorem 8) will apply as there is a causal dependency on the computed gradients, meaning we will be able to write the system as a column-echelon matrix.
>
> Q5. **Non-Gaussian noises**: Our formalism does apply to the Laplace noise mechanism. However, the privacy analysis performed for Theorem 7 (now Theorem 8) would need to be applied to linear combinations of Laplacians instead of a simpler Gaussian random variable, yielding looser bounds. We found Gaussian noises to be more standard and easier to manipulate using $\mu$-GDP.
>
> Q6. **Heterogeneous privacy budget**: Yes, MAFALDA-SGD can be extended to heterogeneous privacy budgets. Algorithm 2 is independent of the privacy budget $\epsilon$, so once the correlation matrix $C_{\text{mafalda}}$ is computed, each node $i$ can target a personalized $(\epsilon_i,\delta_i)$-DP by applying a specific matching scaling factor $\sigma_i$ (Theorem 7 (now Theorem 8)) rather than a global uniform one. An adapted optimization objective that takes into account this heterogeneity is a promising avenue for future work, but falls outside our LDP scenario.
>
> Q7. **Changing participation patterns**: Currently, we only perform the matrix factorization offline, as this is the standard in the online learning litterature. The participation pattern needs to be known to be able to accurately correlate noises. Correlation restarts may be useful in scenarios where the participation pattern can be foreseen only a few steps in advance.

---

> > ### Comment · Reviewer_fuoh · 2025-11-26
> >
> > Thank you for the response. It address my concerns. I will keep my score unchanged.

---

### Official Review · Reviewer_6ixT · 2025-11-01

**Soundness:** 3
**Presentation:** 3
**Contribution:** 3
**Rating:** 6
**Confidence:** 2

**Summary:**

The paper builds a unified privacy-accounting framework for decentralized learning (DL) by recasting DL updates as a Matrix Factorization (MF) mechanism. The paper extends centralized MF-style DP accounting to decentralized settings and multiple trust models. This achives tighter guarantees (especially under PNDP) and motivates a new algorithm, Mafalda-SGD, with optimized, node-wise correlated noise that outperforms prior DP-DL baselines on synthetic and real graphs.

**Strengths:**

1. The paper derives neat theory with clean abstraction and provides a unified lens and extends to adaptive and rectangular cases.
2. The paper presents substantially tighter bounds on PNDP.
3. The paper presents a concrete algorithm Mafalda-SGD that is adoptable and easy to run after an offline correlation evaluation step.
4. Empirically results show strong performance across a lot of graphs.

**Weaknesses:**

1. Seemingly strong assumptions (e.g., linear DL (definition 3), column-echelon, and the knowledge of gossip matrix for MAFALDA-SGD)
2. Limited experimental scale
3. Computational overhead is not discussed

Please see the questions section below for more details.

**Questions:**

1. The theory assumes linear DL where "all observable quantities can be expressed as a linear combination of the concatenated gradients G and the concatenated noise Z." This is often not true in practice. For example, in practice, quantization or compression is often used when exchanging information, which would render them non-linear. Also, if cryptographic techniques are further used to encrypt the weights or to perform secure aggregation, the weights/gradients will usually be converted into a discrete space.
2. In practice, how do we verify/enforce the column-echelon condition on real DL workloads?
3. A client needs to know the gossip matrix and the participation pattern to optimize for correlation. Is this a standard assumption? I guess a client may just know its local patterns. Also the topology may be time-varying.
3. What’s the computational/communication overhead of computing and refreshing the optimal correlation on large graphs and long horizons? How sensitive is the performance to approximate solutions if optimization is hard.
4. While experiments are conducted on a wide range of graphs (both synthetic and real-world), they are limited to small tabular datasets with small models. How would the proposed method for vision or NLP tasks?
5. The paper seems to use GDP and Renyi terms mostly, I recommend the authors to also include (epsilon,delta)-DP guarantees that will be of the interest to many practitioners.

---

> ### Author Response · Authors · 2025-11-20
>
> We thank the reviewer for their careful review. We have uploaded a revised version of the paper with changes highlighted in red and address the questions they raised below.
>
> Q1. **Linear DL concerns**: Non-linear operations are equivalent to a post processing of the full observations. Thus, the privacy bounds derived in this work still hold. Our theory would, however, not account for the possible privacy gains of such non-linear transforms. Moreover, the notion of optimality used by MAFALDA-SGD will be changed by this non-linearity, meaning that the factorization should be adapted to such scenarios. Such scenarios represent a promising venue for future works.
>
> Q2. **Column echelon concerns**: In practice, this is enforced by causality in real DL workload: gradients at time $t$ only depend on earlier gradients. We extend the paragraph after Theorem 7 (now Theorem 8) to reflect this.
>
> Q3. **Knowledge of the communication matrix**: This assumption is necessary to define an optimal correlation that is adapted to the network topology. This is standards in smaller-scale scenarios such as cross-silo settings, or medical settings where each client is an hospital. Moreover, this correlation matrix can be computed offline before training by a trusted third party, without requiring to share the gossip matrix to each client. Finally, see our global comments regarding time varying topologies, which we also address in Remark 3 in the paper's new version.
>
> Q4. **Computational overhead**: The optimization of $C_\text{mafalda}$ (Algorithm 2) is performed over a $T\times T$ matrix, as in the literature on (centralized) optimal MF computation. Thus, the complexity is the same as the one of (Denisov et al, 2022): $O(T^3)$ per steps of L-BFGS. The extra complexity comes from the computation of the matrix $L$ in Algorithm 2. For an arbitrary gossip matrix $W$, the naive computation cost is $O(n^3T+n^2T^3)$, but we expect this cost can be signficantly decreased with efficient matrix representations (e.g., exploiting the sparsity of $W$) and using correlation restarts (Pillutla et al., 2025) to keep $T$ reasonably small. We also note that in typical cross-silo learning (e.g. hospitals using private health data to construct diagnostic models), $n$ will range from a few ten to a few hundred nodes. We added Remark 13 in the paper to discuss the complexity.
>
> We leave the study of sensitivity to approximate solutions to future works. Considering such approximation work well in centralized and federated settings, we infer this should also translate to decentralized settings.
>
>
> Q5. **Additional experiments**: We provide additional experiments for image classification on the FEMNIST dataset in Section 7. The correlation matrix itself does not depend on the number of parameters, and thus the the complexity of the dataset or model does not impact the efficiency of our approach.
>
> Q6. **$(\epsilon, \delta)$-DP**: Following Corollary 1 of Dong et al. (2022), $\mu$-GDP directly and tightly translates into $(\epsilon, \delta)$-DP, with appropriate $\epsilon$ and $\delta$ values. Moreover, it is also possible to derive $(\alpha, \epsilon)$-Rényi DP (Gil et al. (2013)). We add Remark 10 to reflect this.

---

> > ### Comment · Reviewer_6ixT · 2025-11-22
> >
> > I hereby thank the authors for their response and clarification. I will keep my positive score.

---

> > > ### Author Response · Authors · 2025-11-24
> > >
> > > Thank you for your kind feedback.
> > >
> > > If you feel that our response and revised paper have addressed your concerns, we would be grateful if you could consider updating your score. If not, could you please suggest what further steps we could take to improve our paper?

---

### Official Review · Reviewer_dv87 · 2025-11-01

**Soundness:** 3
**Presentation:** 2
**Contribution:** 2
**Rating:** 6
**Confidence:** 3

**Summary:**

This is an interesting paper that addresses differentially private (DP) decentralized learning (DL) by generalizing recent advances in centralized DP. In particular, it achieves privacy be means of correlated noise through the matrix factorization mechanism which is shown to apply also in decentralized settings. The proposed framework is applicable for different trust models, e.g., Local DP, Pairwise Network DP, and Secret-based LDP, providing tighter privacy guarantees and covering all known DP-DL algorithms under a single formulation.

**Strengths:**

- Extends the centralized matrix factorization mechanism to decentralized learning, unifying several existing trust models (LDP, PNDP, SecLDP) and encompassing all known DP-DL algorithms within one framework.

- Generalizes DP analysis to adaptive and non-square workload matrices through Theorem 7, enabling rigorous privacy guarantees for realistic decentralized and potentially other distributed systems.

- Introduces a novel algorithm (MAFALDA-SGD) that optimizes local noise correlation, achieving tighter privacy accounting and better privacy–utility trade-offs than previous methods.

- Empirically validated on multiple graph topologies and datasets, showing consistent performance gains and more accurate privacy bounds compared with prior accounting techniques.

**Weaknesses:**

- The experimental section only include small graphs (the largest including only 271 nodes). Moreover, the models considered in section 7.2 is very small, an MLP with a single hidden layer.

- There are no ablations on the effect of graph topology, participation patterns, or colluding attackers.

- There is no complexity analysis of the new algorithm.

- It is not clear how restrictive definition 3 is.

**Questions:**

1) How restrictive is Definition 3? For example, it seems to roll out adaptive optimization like ADAM, quantization techniques, personalization (often achieved via regularization) etc.

2) In section 6, when defining MAFALDA-SGD, the authors assume that all nodes follow the same local participation pattern. This assumption seem to rule out dynamic topologies that are, in fact, prevalent in DL settings. For example within potential space applications or where nodes may exhibit dropout.
a) Is this assumption necessary for MAFALDA?
b) How does this assumption limit the applicability of MAFALDA?
c) Is the framework applicable to dynamic graphs?

3) Can the authors provide an analysis of the complexity of the approach? Is the framework applicable in realistic scenarios?

4) Algorithm 1 and 2 requires that each node has access to $W$, $\Delta_g$, and $\sigma$.
a) Why is it reasonable for each node to have access to the entire topology?
b) How would $\Delta_g$ be decided in practice? If all nodes must use the same value, it requires coordination that will add to the complexity. How does data heterogeneity among nodes play a role?

5) It would be interesting to see more holistic experiments including i) more realistic settings with larger models, ii) the effect of graph topology, and larger graphs.

---

> ### Author Response · Authors · 2025-11-20
>
> We thank the reviewer for their detailed review. We have uploaded a revised version of the paper with changes highlighted in red and address their questions below.
>
> Q1. **Limitations of Definition 3**: This definition already encompasses all DP-DL algorithms introduced in prior work. We agree that extending it to a wider range of optimizers is a relevant direction, but it is largely orthogonal to our contributions: such extensions are not specific to decentralized learning, and advances developed for centralized MF would likely carry over to our setting. For example, incorporating momentum is straightforward by adjusting the workload matrix. It is also worth noting that vanilla DP-SGD is often difficult to outperform with more complex optimizers such as Adam [1], due to the bias introduced by noise injection. For these reasons, we do not view this as a significant limitation of our work.
> Quantization can be seen as post-processing of the full gradient information. While there may be privacy gains to it, we leave tighter analysis in such scenarios for future work.
> Personalization already falls within the bounds of our current definitions, and can be reflected in the loss used to compute the gradients. While we won't get the same model at the end, the privacy will still hold due to gradient clipping.
>
>
> Q2. **Time-varying graphs**: Our framework supports dynamic graphs (question c), and we added Remark 3 in the paper to clarify this. Regarding dropout (a, b), the formalism of Theorem 7 (now 8) can incorporate it through the sensitivity definition. For MAFALDA-SGD, we assumed identical node participation patterns solely to reduce the space of possible factorizations. Exploring algorithms tailored to heterogeneous local constraints is part of future work, since our primary goal here is to establish privacy guarantees for algorithms expressed via matrix factorization.
>
> Q3. **Complexity of the approach:** The optimization of $C_\text{mafalda}$ (Algorithm 2) is performed over a $T\times T$ matrix, as in the literature on (centralized) optimal MF computation. Thus, the complexity is the same as the one of (Denisov et al, 2022): $O(T^3)$ per steps of L-BFGS. The extra complexity comes from the computation of the matrix $L$ in Algorithm 2. For an arbitrary gossip matrix $W$, the naive computation cost is $O(n^3T+n^2T^3)$, but we expect this cost can be signficantly decreased with efficient matrix representations (e.g., exploiting the sparsity of $W$) and using correlation restarts (Pillutla et al., 2025) to keep $T$ reasonably small. We also note that in typical cross-silo learning (e.g. hospitals using private health data to construct diagnostic models), $n$ will range from a few ten to a few hundred nodes. We added Remark 13 in the paper to discuss the complexity.
>
> Q4. **Hyperparameters access**: For Algorithm 1, each node only needs to know its own neighborhood $\Gamma_u$, as in standard gossip methods. While Algorithm 2 requires the knowledge of the full communication matrix, this is already required for privacy accounting in prior work on PNDP and Sec-LDP, so it is not a limitation specific to our method.
> Regarding $\Delta_g$ (b), it is a clipping norm treated as a standard hyperparameter. As with the model architecture and the correlation matrix $C_\text{mafalda}$, all nodes must share its value, but we do not view this coordination as an unusual overhead. Data heterogeneity does not affect this requirement: $\Delta_g$ is needed universally to compute valid DP guarantees.
>
>
> Q5. **Additional experiments**: We have included experiments on an additional dataset (FEMNIST image classification task) in Section 5 (see in particular Figure 4), with more experiments in the appendix.
> Regarding the number of nodes, we note that existing DL literature typically considers at most a hundred nodes in experiments [2]. This said, our approach can scale to a large number of nodes by using correlation restarts to keep $T$ reasonably small (see Q3 above). To illustrate this, we have added experimental results on the larger Misskey graph [3] ($n=918$) in Figure 4 (now 5). In this case, we are able to efficiently compute the optimal correlation for $(k,b)=(10,16)$ (the main constraint is in the memory requirements).
>
>
> [1] On Design Principles for Private Adaptive Optimizers - Arun Ganesh, Brendan McMahan, Abhradeep Thakurta. arXiv:2507.01129 (2025)
>
> [2] FLamby: Datasets and Benchmarks for Cross-Silo Federated Learning in Realistic Healthcare Settings - Ogier du Terrail et al. NeurIPS 2022
>
> [3] Damie, Marc, and Edwige Cyffers. "Fedivertex: a Graph Dataset based on Decentralized Social Networks for Trustworthy Machine Learning." arXiv preprint arXiv:2505.20882 (2025).

---

> > ### Comment · Reviewer_dv87 · 2025-11-20
> >
> > Thanks for the response. I will keep my positive score.

---

> > > ### Author Response · Authors · 2025-11-24
> > >
> > > Thank you for your kind feedback.
> > >
> > > If you feel that our response and revised paper have addressed your concerns, we would be grateful if you could consider updating your score. If not, could you please suggest what further steps we could take to improve our paper?

---

### Author Response · Authors · 2025-11-20
**Global Comment**

We thank all reviewers for their insights and feedback. We provide point-by-point answers to each reviewer below. We have also uploaded a revised version of the paper with changes indicated in red, and highlight here how we addressed two questions raised by multiple reviewers:
* **Additional dataset**: Reviewers (dv87, fuoh, 6ixT) expressed their interest in seing experiments on more diverse learning tasks. We added image-classification experiments on Federated EMNIST (FEMNIST) with a ConvNet. The results confirm the superiority of MAFALDA, in line with our previous experiments (regression on tabular data). These experiments are reported in Section 7 (Figure 4) and Appendix C.2.
* **Time-varying communication graphs**: Reviewers (dv87, 6ixT, fuoh) asked whether our framework supports time-varying communication matrices $W_t$. We have clarified that the formalism, the theoretical results, and MAFALDA all natively accommodate this time-varying setting. This may have been unclear because Section 4.1 focuses on the constant-matrix case for pedagogical clarity. To remove any remaining ambiguity, we have added Remark 3 and Appendix A.7, which explicitly detail how the general time-varying case is handled.

---

### Author Response · Authors · 2025-12-01
**Summary of the Discussion Phase**

Dear AC,

We thank you for handling the difficult situation caused by the data leak. To wrap up the discussion process, we provide a summary below.

First, we thank all reviewers for their positive feedback, comments and suggestions, which allowed us to substantially improve our work's clarity and structure, as well as strengthen its experimental results. In particular, following the reviewers' suggestions:
1. We added experiments on a larger dataset, with a larger model and a larger graph.
2. We clarified that our framework applies to broader contexts, such as time-varying communication matrices.

To the best of our knowledge, our rebuttal addressed all reviewers' concerns: three reviewers explicitly indicated their satisfaction, and no additional issues were raised after we submitted the rebuttal and revised version on November 20th.

We would like to highlight the main strength and novelty of this paper again: a unified framework for privacy-preserving decentralized learning that enables automatic, tight privacy-budget computation across multiple threat models and algorithms, including approaches with correlated noise that are typically hard to analyze. The work also introduces a new family of algorithms that can serve as a foundation for future explorations of alternative noise-correlation strategies in decentralized learning, and potentially new notions of optimality.

For these reasons, we hope that the AC will recommend acceptance, and perhaps consider the paper for spotlight.

Thank you for your consideration and flexibility in this exceptional and difficult review context.

The Authors.

---

### Meta-Review · Area_Chair_doDu · 2026-01-07

**Summary:**

This paper introduces a unified matrix factorization framework for differentially private decentralized learning, yielding tighter privacy guarantees and the MAFALDA-SGD algorithm. It successfully bridges centralized privacy accounting with decentralized constraints, offering a versatile theoretical tool for future research.

**Reviewer Concerns:**

The rebuttal effectively resolved major concerns regarding experimental diversity (FEMNIST added) and support for dynamic topologies. While the offline optimization overhead is non-negligible, it is justified within the context of cross-silo learning scenarios targeted by the work.

**Reviewer Scores:**

All reviewers maintained a score of 6, reflecting a consensus that the work is technically sound and above the acceptance threshold. Had the reviewers fully appreciated the long-term impact of the generalized theoretical framework beyond the specific MAFALDA instantiation, scores likely would have been higher.

Reasons for Acceptance

Unified Framework: Establishes a novel, rigorous theoretical unification of diverse trust models (LDP, PNDP, SecLDP) under a single matrix factorization umbrella.

Tighter Accounting: Provides provably tighter privacy guarantees than existing baselines, enabling better utility-privacy trade-offs in decentralized settings.

Robust Rebuttal: The addition of image classification tasks (FEMNIST) and larger graph experiments successfully demonstrated the method's broader applicability and robustness.

Algorithmic Novelty: MAFALDA-SGD introduces a practical mechanism for correlated noise injection, serving as a foundational baseline for future privacy-preserving decentralized algorithms.

---

### Decision · Program_Chairs · 2026-01-26

Accept (Poster)